# Effectiveness of a community-based intervention package in maternal health service utilization: A cross-sectional quasi-experimental study in rural Ghana

Hirotsugu Aiga[1,2,3]*, Yoshito Kawakatsu[4], Nobuhiro Kadoi[3], Emmanuel Obeng[5], Frank Tabi Addai[5], Frederick Ofosu[6], Kazuki Fujishima[7], Mayumi Omachi[8], Etsuko Yamaguchi[7]

1 School of Tropical Medicine and Global Health, Nagasaki University, Nagasaki, Japan, 2 Institute of Tropical Medicine (NEKKEN), Nagasaki University, Nagasaki, Japan, 3 Human Development Department, Japan International Cooperation Agency (JICA), Tokyo, Japan, 4 Department of Nursing, Graduate School of Biomedical Sciences, Nagasaki University, Nagasaki, Japan, 5 Ghana Country Office, The Japanese Organization for International Cooperation in Family Planning (JOICFP), Suhum, Eastern Region, Ghana, 6 Suhum Municipal Health Directorate, Ghana Health Service (GHS), Suhum, Eastern Region, Ghana, 7 International Program, The Japanese Organization for International Cooperation in Family Planning (JOICFP), Tokyo, Japan, 8 Interfaculty Initiative in Planetary Health, Nagasaki University, Nagasaki, Japan

* hirotsuguaiga@gmail.com, hirotsugu_aiga@yahoo.co.jp, hiroaiga@gwu.edu

## Abstract

### Background

We examined the effectiveness of a community-based intervention package that targeted pregnant women for increasing utilization of maternal health services. The intervention package was implemented in Suhum Municipality, Ghana, from March 2019 to April 2022. The package consisted of: (i) maternal health education by female and male peers; (ii) training existing health workers on maternal health; and (iii) strengthening the local community health management committees.

### Methods

A cross-sectional household survey was conducted in May 2022. We sampled four groups of women: (i) intervention at baseline; (ii) intervention at follow-up; (iii) control at baseline; and (iv) control at follow-up. Three outcome variables were set, i.e., the proportions of women having utilized: (i) at least four antenatal care (ANC) services; (ii) facility-based delivery (FBD) services; and (iii) post-partum care (PPC) services. To estimate the effectiveness of the intervention package in increasing the service coverages, both crude and adjusted difference-in-differences (DID) estimates were calculated. Significance levels were set at the values of 0.10, 0.05, and 0.01, since the aforementioned service coverages had already been too high to have room for an increase at the significance level of 0.05.

**Data Availability Statement:** All relevant data are within the manuscript and its Supporting Information files.

**Funding:** This study is funded by Takeda Global CSR Program of Takeda Pharmaceutical Company Limited. The funder had no role in study design, data collection and analysis, decision to publish, or and preparation of the manuscript.

**Competing interests:** EO, FTA, KF and EY were partially engaged either in planning or in implementing the Program. All the other authors (HA, YK, NK, FO and MO) declare that they have no competing interests. This does not alter our adherence to PLOS ONE policies on sharing data and materials. There are no patents, products in development or marketed products associated with this research to declare. Authors' acronyms: Hirotsugu Aiga (HA); Yoshito Kawakatsu (YK); Nobuhiro Kadoi (NK); Emmanuel Obeng (EO); Frank Tabi Addai (FTA); Frederick Ofosu (FO); Kazuki Fujishima (KF); Mayumi Omachi (MO); and Etsuko Yamaguchi (EY).

**Abbreviations:** ANC, Antenatal care; ANC4, At least four antenatal care visits; aOR, adjusted odd ratio; CHPS, Community-based Health Planning and Services; CHVs, Community health volunteers; CI, Confidence interval; COVID-19, Coronavirus Disease 2019; DID, Difference-in-differences; FBD, Facility-based delivery; GHS, Ghana Health Service; JOICFP, Japanese Organization for International Cooperation in Family Planning; NHIS, National Health Insurance Scheme; OR, Odds ratio; PPC, Postpartum care; TDRC, Tropical Diseases Research Centre; TMGH, School of Tropical Medicine & Global Health, Nagasaki University; UNFPA, United Nations Population Fund; WHO, World Health Organization.

## Results

The proportion of women completing at least four ANC services displayed significant DID in both crude and adjusted analyses. The proportions of women utilizing facility-based delivery services and post-partum care services did not display significant DID.

## Conclusions

Of the three outcome variables set, only the proportion of women having utilized at least four ANC services significantly increased in the intervention group, compared with the control group. Ghana has been in the transition process of shifting the minimum number of ANC visits from four to eight. Thus, nationwide scaling up of the intervention package is expected to help the transition be smooth by increasing the number of ANC visits.

## Background

Today, approximately 800 women die due to the preventable causes of maternal deaths in the world, every day [1]. Preventable causes of maternal deaths are composed of direct causes (obstetric hemorrhage, hypertensive disorders, puerperal sepsis, and unsafe abortion) and indirect causes (HIV and AIDS, malaria, tuberculosis, other sexually transmitted infections, malnutrition) [2]. They can be prevented through increasing availability of and access to quality of sexual, reproductive, and maternal care. More than 60% of maternal deaths occur in Sub-Saharan Africa [3]. Nearly 75% of maternal deaths are caused by severe bleeding, infections, pre-eclampsia and eclampsia, complications from delivery, and unsafe abortion whose majority cases are either preventable or treatable [4]. To reduce a significantly greater number of maternal deaths in Sub-Saharan Africa, timely and quality antenatal care (ANC) and facility-based delivery (FBD), post-partum care (PPC), Emergency Obstetric Care, and family planning are imperative [5–7]. Poorer access to health facilities, inadequate quality of health services, and socio-economic and socio-cultural barriers are the major obstacles in increasing maternal health service utilization. It prevents reproductive-aged women from reaching life-saving services [8, 9].

Improvement of women's access to health facilities and health workers alone does not increase the reproductive-aged women's utilization of the aforementioned key maternal health services, for instance by increasing the number of health facilities and availability of public transports. Creation of health service demands through direct intervention on women in communities needs to be undertaken in parallel [10–12]. Health service demands can be created commonly through household visits and health education sessions conducted by facility-based and/or community-based formal health workers [13, 14]. However, demand creation initiated and implemented by formal health workers does not necessarily result in producing the expected effectiveness. This is largely because the formal health workers' way of demand creation is not necessarily interactive and friendly enough for local women [15–17]. On the other hand, peer education is expected to play a vital role in increasing service utilization, as the demands can be created more from service users' perspective. Peer education has been often employed as an effective approach for increasing service utilization not only in adolescent health [18], family planning [19, 20], HIV and other sexually transmitted infection control [21–25], but also in maternal health [10, 26]. Moreover, the interventions based on locally

available resources (e.g. peer educators) are more likely to be sustainable by being integrated into local community systems, than facility-based interventions.

To improve reproductive and maternal health among reproductive-aged women (both adults and adolescents) in four African countries (i.e., Ghana, Kenya, Tanzania and Zambia), the Japanese Organization for International Cooperation in Family Planning (JOICFP) implemented a five-year program '*Protecting the Lives of Pregnant Women in Africa*: *Community-centered sustainable health promotion program*' (the Program) during the period from January 2018 to December 2022 [27]. The Program was aimed at increasing pregnant women's utilization of three key maternal health services (i.e., ANC, FBD, and PPC). The Program employed a community-centered approach to ensure its sustainable effectiveness beyond the program life. Both female and male peers recruited from local adults and youths through a series of consultations with the community key stakeholders, were trained as community health volunteers (CHVs) responsible for peer educations and peer consultations on reproductive and maternal health. They were responsible also for locally disseminating the information related to reproductive and maternal health and rights to their peers, and referring those in need of reproductive and maternal health care to primary health facilities. The Program also strengthened the functions and capacity of the existing community health management committees, to enable them to continuously support both CHVs and facility-based health workers. Capacity strengthening of community health management committees was conducted through a series of lectures, group works, and role plays. The Program trained health workers at primary health facilities to improve quality of facility-based health services, too.

By the end of December 2019, the JOICFP in collaboration with its local government partners of the four countries trained 1,856 CHVs composed of males and females, and adults and youths. Through the CHVs, a total of 466,549 individuals were reached in the Program sites with the promotive messages on reproductive and maternal health as of 31st March 2021. Of them, 79,662 were referred to primary health facilities.

To estimate the effectiveness of the aforementioned intervention package in reproductive and maternal health, this study was conducted in Ghana. There were three reasons for conducting the study only in Ghana. First, as the implementation of the Program was less affected by the SARS-CoV-2 (COVID-19) pandemic in Ghana than in other three countries. This study was conducted to estimate the effectiveness of the aforementioned intervention package in reproductive and maternal health in Ghana,. Second, the largest amount of funds was invested in Ghana (i.e., 45% of the total local implementation costs of all the four target countries of the Program). Third, the activities of the Program were fully implemented only in Suhum Municipality of Ghana throughout the entire five-year Program period. In Ghana, the Community-based Health Planning and Services (CHPS) initiative has been implemented since 2000 after careful piloting [28]. The initiative is aimed at ensuring availability of and access to basic maternal and child health services (antenatal care, delivery, postpartum care, child immunization, growth monitoring, and integrated management of childhood illnesses), in particular consideration to women in rural areas. Two community health officers are assigned at each CHPS compound, a primary health facility, to enable them to provide both facility-based and community-based maternal and child health. Specific objectives of the study are estimations of the degree of increases in proportions of women having utilized: (i) at least four ANC services; (ii) FBD services; and (iii) PNC services. National Health Insurance Scheme (NHIS) covers the significant part of maternal and child health service costs for those insured. Yet, not all costs are covered by NHIS and, needless to say, out-of-pocket payments made by those not insured become much larger. Thus, the poor have greater challenges in utilization of health services. The findings of the study are expected not simply to estimate the

effectiveness of the Program, but also to serve as the key evidence for improving the design of the Program in the future.

## Methods

A quasi-experimental study was conducted in Ghana. Of the four countries where the Program was implemented, Ghana was targeted for the study. Note that the community-based approach similar to this intervention package was previously tested in several countries by the JOICFP [29–31]. Their crude effectiveness was estimated in a cross-sectional observational study, while it has not been precisely assessed [32].

### Study design

To estimate the effectiveness of the Program in reproductive and maternal health care service utilization, difference-in-differences (DID) approach was employed [33]. A cross-sectional household survey was conducted, to compare reproductive and maternal health care service utilization between intervention and control sites. In the survey, we sampled four types of groups of women who gave live births during pre-intervention (baseline) and post-intervention (follow-up) periods for both intervention and control groups: i.e., Group $I_1$ as the intervention group at baseline, Group $I_2$ as the intervention group at follow-up, Group $C_1$ as the control group at baseline, and Group $C_2$ as the control group at follow-up (Fig 1).

### Study area

Of seven municipalities/districts intervened by the Program in Eastern Region, Suhum Municipality was selected as the intervention site for this study. This is because Suhum Municipality was the only municipality/district where the Program implementation was long and deep enough (i.e., >2 years) to assess its effectiveness. Note that the Program intervention, nevertheless, was suspended during the period from March to October 2020 due to COVID-19 pandemic in all the intervention municipalities/districts of Ghana (incl. Suhum Municipality). The other six were the municipalities/districts where the interventions of the Program started later than in Suhum Municipality (i.e., Akyemansa District, Birim North District, Kwahu East District, Lower Manya Krobo Municipality, Upper Manya Krobo District, and Yilo Krobo Municipality). Atiwa West District was selected as the control site where the Program's intervention was neither implemented nor planned at the time of the study. Atiwa West District was appropriate as the control group also because its socio-economic and socio-demographic characteristics, and maternal health service coverages at the baseline were reportedly at the similar level to those of Suhum Municipality. Another reason for selecting Atiwa West District is that no externally supported maternal and/or reproductive health project had been planned and implemented in the district. In both Suhum Municipality (intervention site) and Atiwa West District (control site), all the basic health services described earlier were made available.

### Study participants

All the women who delivered between 1st January 2018 and 28th February 2019 before the Program's launch in March 2019 were included in the baseline sampling frames for both intervention and control groups (Group $I_1$ and Group $C_1$ in Fig 1). Similarly, all the women who delivered between 1st August 2021 and 30th April 2022 after the 10 months had passed since the Program's launch were included in the follow-up sampling frames for both intervention and control groups (Group $I_2$ and Group $C_2$ in Fig 1). Those not having continued to live in Suhum Municipality during the entire pregnancy were excluded from the follow-up sampling

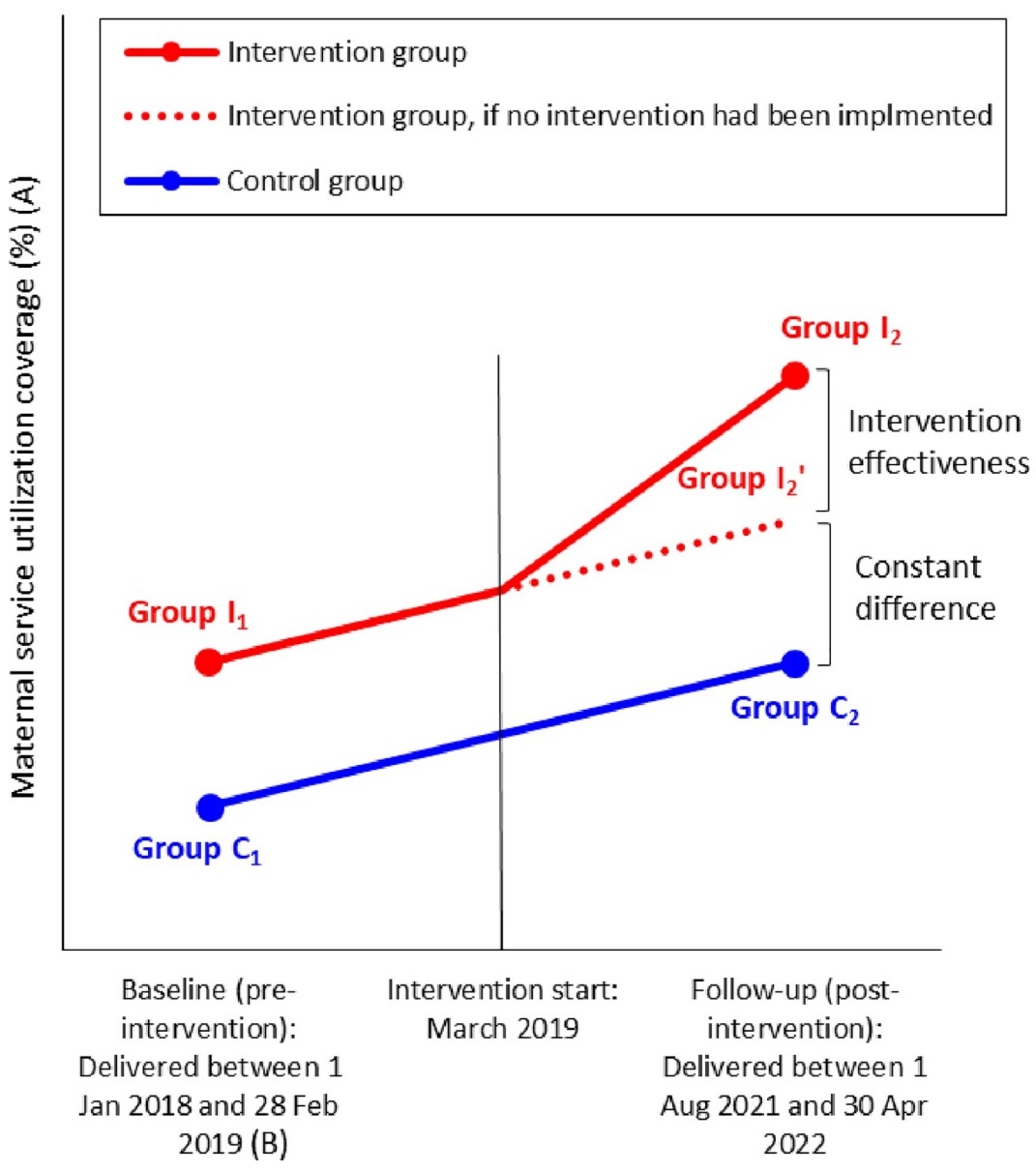

**Fig 1. Difference-in-differences framework for this study.** (A) Three types of maternal health service coverages were employed: (i) **ANC:** at least four antenatal care visits as proxy for continuation of ANC; (ii) **FBD:** facility-based delivery; and (iii) **PPC:** at least one postpartum care visit within two days after delivery. (B) Utilizations of the three types of maternal health services were measured, based on mothers' memories at the time of the cross-sectional survey.

frame for the intervention group (Group $I_2$ in Fig 1). By doing so, we ensured that, in the intervention group, the follow-up study participants were fully exposed to the intervention during their pregnancies, while the baseline participants were never. Those having been relocated from Suhum Municipality and other six intervention municipalities/districts to Atiwa West District during their pregnancies were excluded from the follow-up sampling frame for the control group (Group $C_2$ in Fig 1). By doing so, we ensured that, in the control group, the follow-up study participants were never, even partially, exposed to the intervention during their pregnancies.

### Interventions and outcome variables

The Program intervention package was composed of three pillars: (i) peer health education and promotion on reproductive and maternal health by locally recruited CHVs; (ii) improvement of the quality of reproductive and maternal health care by training existing health workers at primary health facilities on reproductive and maternal health; and (iii) strengthening of the local community health management committees (Fig 2).

**1st pillar: Social and behavior change communication**
- Locally recruit and train CHVs
- Develop communication strategies, tools and messages
- Conduct peer health education and promotion activities for demand creation
- Refer women in need of reproductive and maternal care to local primary health facilities

**2nd pillar: Quality reproductive and maternal health services**
- Train facility-based health workers on client-centered care (respectful maternal health care, adolescent-friendly services)
- Implement 5S-initiative (*Sorting, Setting, Shining, Standardization, and Sustain*)

**3rd pillar: Community-centered sustainable mechanism**
- Strengthen capacity of community health management committees on evidence-based planning, implementation and monitoring
- Provide CHVs with supportive supervision
- Undertake local resource mobilization for community-based health activities

**Fig 2. Design of the intervention package.**

**Table 1. Education attainments of two types of community health volunteers.**

|  | Maternal and Child Health Promoters | | Peer Educators | |
|---|---|---|---|---|
|  | *N* | (%) | *n* | (%) |
| No education | 26 | 14.4% | 14 | 9.3% |
| Junior high school | 66 | 36.7% | 56 | 37.3% |
| Senor high school | 53 | 29.4% | 66 | 44.0% |
| University or college | 35 | 19.4% | 14 | 9.3% |
| Total | 180 | 100.0% | 150 | 100.0% |

Under the first pillar, the Program trained a total of 180 adult and 150 youth CHVs (Maternal and Child Health Promoters and Peer Educators, respectively) in Suhum Municipality. As shown in Table 1, while a majority of Peer Educators completed either junior or senior high school education (81.3%), education attainments of Maternal and Child Health Promoters are more diverse.

To effectively create reproductive and maternal health care demands, the trained CHVs conducted peer health education and promotion activities at the community level (i.e., individual/group health education and consultation sessions, and referrals of women in need of maternal health care services to local primary health facilities). The Program provided technical support to participatory development of communication strategies, tools and messages. They were designed so as to address the local common obstacles in accessing maternal and sexual reproductive health services from health service users' perspective.

Under the second pillar, the Program supported the municipal/district health directorates in training facility-based health workers on client-centered care (i.e., respectful maternity care, adolescent-friendly services, and *5S-Initiative* activities). *5S-Initiative* is the conceptual framework composed of *Sorting*, *Setting*, *Shining*, *Standardization*, *and Sustaining* that helps organize and manage the workplace and workflow with the intent to improve efficiency, by eliminating waste, improving flow and reducing process unreasonableness [34].

Under the third pillar, the capacity of community health management committees was strengthened in the implementation of evidence-based planning, implementation, monitoring and evaluation as well as resource mobilization for sustaining the CHVs' activities and health workers' quality client-centered care. Capacity strengthening of community health management committees was conducted through a series of lectures, group works, and role plays. These three pillars are mutually complemented to synergically ensure women's timely access to maternal-health-related information and care.

To estimate the effectiveness of the interventions of the Program, three variables related to the utilization of essential maternal health services (i.e., ANC, FBD and PPC) were set as the outcome variables (Table 2). They were selected as the outcome variables since they serve as the appropriate proxies for health-seeking behavior changes that can be expected to be

**Table 2. Three outcome variables for the study.**

| Outcome variable | | Definition |
|---|---|---|
| **ANC** | At least four antenatal care visits | Proportion of women having utilized antenatal care services provided by qualified health professionals at least four times: (%) |
| **FBD** | Facility-based delivery | Proportion of women having utilized delivery services at public or private health facilities: (%) |
| **PPC** | Postpartum care visit | Proportion of women having utilized postpartum cares within two days after the birth: (%) |

achieved through the interventions of the Program. All the three outcome variables were defined as dichotomous variables (i.e,. 0 as not utilized and 1 as utilized). In addition to the outcome variables, socio-economic and socio-demographic data were collected as the independent variables, by conducting structured interviews in local languages (Akan, Ewe and Krobo).

## Sample size and sampling

The sample size was calculated, so as to detect a significant DID for each outcome variable, by making finite population correction adjustments [35]. This is because the target group was the specific populations in limited geographic areas: i.e., those who delivered between 1st January 2018 and 28th February 2019 (baseline) and those who delivered between 1st August 2021 and 30th April 2022 (follow-up), in Suhum Municipality (intervention site) and Atiwa West District (control site). Those having delivered during the period other than the above two periods were excluded from the sampling. Moreover, those having been relocated, absent despite three repeated visits or refused to participate in the study were excluded, too. Then, design effect of 1.3 was multiplied, as two-stage cluster sampling where mean cluster size was 25 was employed [36]. Assuming 10% non-response rate, the final sample sizes was set at 1,812 mothers (i.e., 453 for each of four groups: Group $I_1$, Group $I_2$, Group $C_1$ and Group $C_2$ in Fig 1). Systematic random sampling was used for selecting mothers having children of respective age groups, for both first and second sampling stages. Expanded Programme on Immunization (EPI) registers readily available at all the Community-based Health Planning and Services (CHPS) compounds (primary healthcare facilities in Ghana) in Suhum Municipality and Atiwa West District were used as sampling frames of mothers of children.

## Data analysis

A series of data analyses were conducted in a two-step manner. First, descriptive analysis was conducted for both outcome/dependent and independent variables. As a part of descriptive analyses, the characteristics of study participants were presented. Second, to estimate the effectiveness of the interventions of the Program, both crude and adjusted DID values were estimated. For calculating adjusted DID estimates, mixed-effect logistic regression was employed, by applying sub-district as the random effect of and independent variables (i.e., group, time, maternal age, education attainment, marital status, enrollment in health insurance, household size, and household's wealth group) as the fixed effects. To test possible multicollinearity, variance inflation factor (VIF) was assessed for each independent variable, by applying VIF value equal to or greater than 10 as the presence of multi-collinearity. Significance levels were set at the values of 0.10, 0.05, and 0.01. The reason for unusually employing 0.10 as one of the significance levels was that the service coverages of the outcome variables had been already too high to have room for an increase at the significance level of 0.05 (e.g., 93.1% of ANC in the intervention group at baseline). Moreover, note that 0.10 has been widely employed as a significance level for behavioral studies into which this study can be categorized [37, 38].

Wealth index was calculated for each household to which respective respondent women belonged, by applying the variables on households' ownership of 11 types of utilities and assets to principal component analysis. The 11 types of utilities and assets were composed of bicycle, cart, electricity, fixed-line telephone, motorbike, radio, personal computer, refrigerator, television set, vehicle, and watch. The principal component score was calculated for each household, by applying all the dichotomous variables of ownership of 11 types of utilities and assets to principal component analysis with varimax rotation [39]. Then, all the households were categorized into three groups (i.e., wealth tertiles) according to the values of wealth index. In

addition, for an additional analysis to assess the impact of interventions by wealth groups, we divided all the households into two groups (i.e., rich and poor) to produce robust estimates and avoid quite high standard errors due to small sample size in each group. All the statistical analyses were conducted, by using Stata 17 (Stata Corp LLC, College Station, TX, USA).

### Ethical considerations

An informed consent to participate in the study was obtained in a written form from each mother after the explanation of the study upon household visits. For mothers under 18 years of age, both informed assent and informed consent were obtained from them and their guardians, respectively. In view of the ongoing COVID-19 pandemic, the interviewers practiced either hand washing or alcohol-based hand sanitization both before and after the interviews, and wore face masks during an interview by keeping themselves 1.5 meters away from the respondent. The authors had access to the information that could identify individual participants during and after data collection. To ensure the confidentiality of personal data of the participants, the dataset for analyses was anonymized by encoding the data with unique participant identification numbers. Another dataset composed exclusively of the identification numbers and participants' personal data (i.e., names and phone numbers) was separately created. Both datasets were protected by using the different passwords.

The study protocol was both internationally and locally approved by: (i) the Institutional Review Board, School of Tropical Medicine Global Health, Nagasaki University, Japan (Approval no. NU_TMGH_2021_179_1); and (ii) Ethics Review Committee, Ghana Health Service, Ghana (Approval no. GHS-ERC 001/02/22). The official permissions for the implementations of the study were obtained from both Suhum Municipal Health Directorate and Atiwa West District Health Directorate. Moreover, local agreements on the implementation of the study were obtained from the leaders of all the target communities, prior to data collection.

## Results

Data collection was conducted during the period from 5th to 29th May 2022. Of 1,812 mothers sampled, 1,471 participated in the study (76.1%). Those not having participated were composed of: (i) 158 having already relocated; (ii) 148 ineligible without meeting the inclusion criteria; (iii) 20 absent; (iv) 14 refusals; and (v) one withdrawal in the middle of the interview.

### Characteristics of participants

Table 3 shows the socio-demographic and socio-economic characteristics of the respondent mothers, by comparing between the intervention and control groups and between baseline and follow-up. Significant differences were unexpectedly detected in the proportions and mean values of nine and seven of 12 characteristic variables between the intervention and control groups at the baseline and follow-up respectively. This indicates that a need for calculating an adjusted DID estimate for each outcome variable by controlling its confounders, in addition to estimating its crude DID. In both groups, approximately 70% or more of mothers completed secondary education or higher. More than three quarters of their households had at least one mobile phone. Approximately, 90% or more of the mothers were either protestants or other types of Christians.

### Overall difference-in-differences

Fig 3(A)–3(C) (show crude DID for three outcome variables, i.e., ANC, FBD and PPC, respectively. Of the three outcome variables, ANC produced a significant crude DID between the

**Table 3. Socio-demographic and socio-economic characteristics of participants.**

| Characteristic variable | Baseline (n = 692) | | | | P-value | Follow-up (n = 639) | | | | P-value |
|---|---|---|---|---|---|---|---|---|---|---|
| | Intervention (n = 393) | | Control (n = 299) | | | Intervention (n = 438) | | Control (n = 340) | | |
| Maternal age [year] (mean, sd) | 30.7 | 7.3 | 31.6 | 6.6 | 0.086* a | 28.6 | 6.8 | 28.6 | 7.0 | 0.890 a |
| Education attainment (n, %) | | | | | 0.016** b | | | | | 0.169 b |
| None / pre-primary education | 21 | 5.3% | 15 | 5.0% | | 23 | 5.3% | 25 | 7.4% | |
| Primary education | 109 | 27.7% | 58 | 19.5% | | 87 | 19.9% | 50 | 14.8% | |
| Secondary and vocational education | 256 | 65.1% | 224 | 75.2% | | 314 | 71.7% | 247 | 73.3% | |
| University / post graduate education | 7 | 1.8% | 1 | 0.3% | | 14 | 3.2% | 15 | 4.5% | |
| Marital status (n, %) | | | | | < 0.001*** b | | | | | < 0.001*** b |
| Single not living with partner | 118 | 30.0% | 27 | 9.0% | | 137 | 31.3% | 40 | 11.8% | |
| Cohabiting living with partner | 170 | 43.3% | 124 | 41.5% | | 166 | 37.9% | 160 | 47.1% | |
| Married living with partner | 89 | 22.6% | 133 | 44.5% | | 107 | 24.4% | 135 | 39.7% | |
| Married not living with partner | 8 | 2.0% | 4 | 1.3% | | 26 | 5.9% | 5 | 1.5% | |
| Separated/widowed/divorced | 8 | 2.0% | 11 | 3.7% | | 2 | 0.5% | 0 | 0% | |
| Enrollment in health insurance (n, %) | | | | | 0.149 b | | | | | 0.001*** b |
| Enrolled | 371 | 94.4% | 290 | 97.0% | | 418 | 95.4% | 339 | 99.7% | |
| Not enrolled/do not know | 22 | 5.6% | 9 | 3.0% | | 20 | 4.6% | 1 | 0.3% | |
| Household size [person/HH] (mean, sd) | 2.33 | 1.28 | 2.53 | 1.28 | 0.048** a | 2.53 | 2.34 | 1.25 | 1.39 | 0.010* a |
| Ownership of household assets (n, %) | | | | | | | | | | |
| Television set | 319 | 81.2% | 247 | 82.6% | 0.699 b | 365 | 83.3% | 280 | 82.4% | 0.792 b |
| Radio | 262 | 66.7% | 200 | 66.9% | 1.000 b | 304 | 69.4% | 236 | 69.4% | 1.000 b |
| Mobile phone | 307 | 78.1% | 267 | 89.3% | < 0.001*** b | 350 | 79.9% | 297 | 87.4% | 0.008*** b |
| Bicycle | 48 | 12.2% | 90 | 30.1% | < 0.001*** b | 47 | 10.7% | 89 | 26.2% | < 0.001*** b |
| Motorbike | 60 | 15.3% | 71 | 23.7% | 0.006*** b | 55 | 12.6% | 67 | 19.7% | 0.009*** b |
| Vehicle | 25 | 6.4% | 40 | 13.4% | 0.003*** b | 30 | 6.8% | 48 | 14.1% | 0.001*** b |
| Religion (n, %) | | | | | 0.001*** b | | | | | 0.472 b |
| Roman catholic | 3 | 0.8% | 6 | 2.0% | | 6 | 1.4% | 10 | 2.9% | |
| Protestant and other Christianity | 373 | 94.9% | 260 | 87.0% | | 397 | 90.6% | 303 | 89.1% | |
| Muslim | 10 | 2.5% | 28 | 9.4% | | 27 | 6.2% | 22 | 6.5% | |
| Other religion and no religion | 7 | 1.8% | 5 | 1.7% | | 8 | 1.8% | 5 | 1.5% | |

* $P < 0.10$

** $P < 0.05$

*** $P < 0.01$

a One-way analysis of variance (ANOVA)

b Chi-square test

intervention and control groups while the others did not. Note that these crude DID values are likely to be less accurate because they were estimated without controlling the possible confounders despite difference in characteristics of mothers between the two groups.

Therefore, adjusted DID estimates were calculated by controlling the possible confounders, using mixed-effect logistic regressions. None of independent variables produced VIF values equal to or greater than 10. Thus, no multicollinearity was assumed. Table 4 shows the results of mixed-effect logistic regressions for the three outcome variables. Similarly to crude DID, only ANC produced a significant adjusted DID estimate (aOR = 2.23; $P = 0.099 < 0.10$) also in the regression models. FBD and PPD did not produced significant adjusted DID estimates. Thus, significant effectiveness of the intervention package among mothers was detected exclusively for ANC.

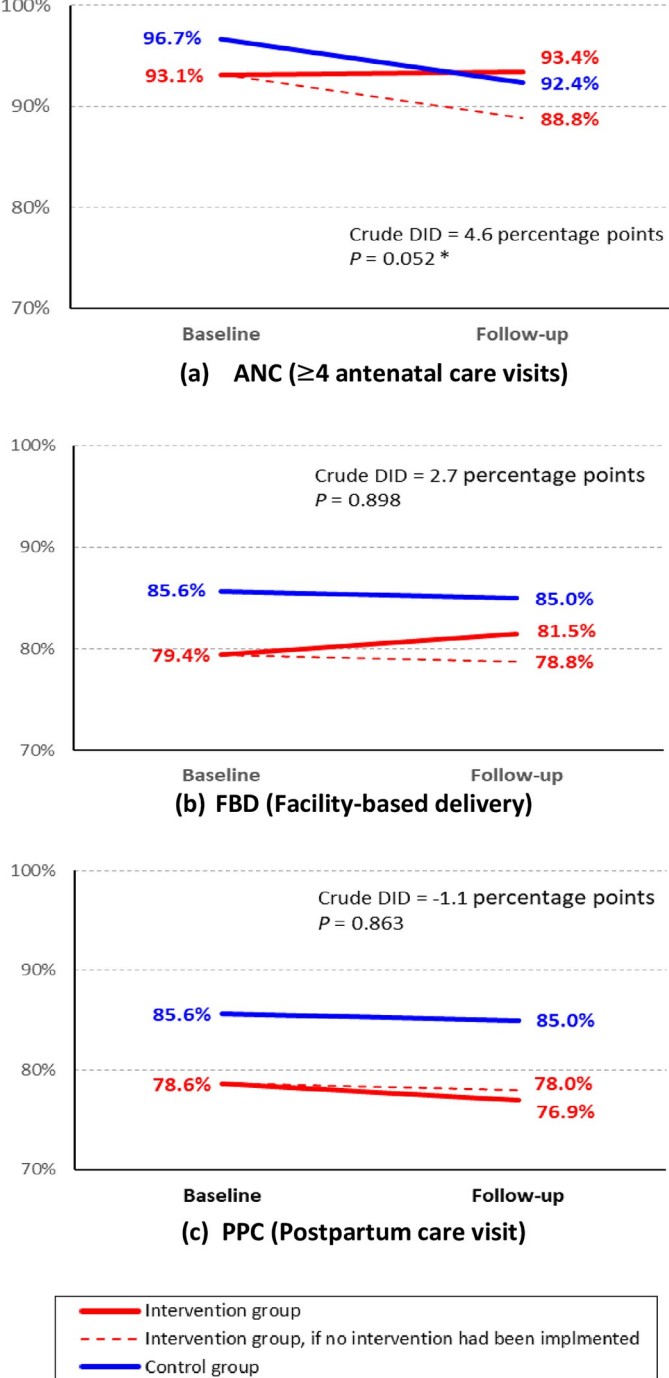

**Fig 3. Overall crude difference-in-differences for three outcome variables.** Crude DID is expressed in form of percentage points. * $P < 0.10$, ** $P < 0.05$, *** $P < 0.01$.

## Wealth-group-specific difference-in-differences

As shown in Table 4, the household wealth group was the only independent variable that systematically produced significant odds ratio across the three models. This implies that the household wealth group was a common predictor of effectiveness of the intervention package.

**Table 4. Results of mixed-effect logistic regressions for three outcome variables: Overall analysis.**

| Independent variable | Model 1 | | | Model 2 | | | Model 3 | | |
|---|---|---|---|---|---|---|---|---|---|
| | ANC: ≥4 Antenatal care visits | | | FBD: Facility-based delivery | | | PPC: Postpartum care visit | | |
| | aOR | 95% CI | P-value | aOR | 95% CI | P-value | aOR | 95% CI | P-value |
| Intercept | 3.12 | 0.79–12.32 | 0.105 | 1.71 | 0.57–5.07 | 0.337 | 1.95 | 0.69–5.46 | 0.205 |
| Group | | | | | | | | | |
| Control | (Ref.) | | | (Ref.) | | | (Ref.) | | |
| Intervention | 0.71 | 0.30–1.68 | 0.437 | 0.70 | 0.34–1.41 | 0.318 | 0.80 | 0.44–1.46 | 0.465 |
| Time | | | | | | | | | |
| Baseline | (Ref.) | | | (Ref.) | | | (Ref.) | | |
| Follow-up | 0.48* | 0.22–1.04 | 0.064 | 1.21 | 0.75–1.95 | 0.434 | 1.03 | 0.65–1.65 | 0.887 |
| Adjusted DID estimates [a] | 2.23* | 0.86–5.77 | 0.099 | 0.95 | 0.53–1.73 | 0.875 | 0.91 | 0.51–1.61 | 0.739 |
| Maternal age [year] | | | | | | | | | |
| < 20 | (Ref.) | | | (Ref.) | | | (Ref.) | | |
| 20–29 | 1.48 | 0.70–3.10 | 0.302 | 1.49 | 0.84–2.65 | 0.172 | 1.02 | 0.58–1.81 | 0.933 |
| 30–39 | 2.06 | 0.93–4.55 | 0.073* | 1.76 | 0.97–3.20 | 0.061* | 1.39 | 0.77–2.51 | 0.275 |
| ≥40 | 1.52 | 0.58–4.03 | 0.398 | 1.41 | 0.70–2.83 | 0.334 | 1.07 | 0.54–2.14 | 0.847 |
| Education attainment | | | | | | | | | |
| None / pre-primary education and primary education | (Ref.) | | | (Ref.) | | | (Ref.) | | |
| Secondary and higher education | 1.22 | 0.76–1.96 | 0.420 | 1.80 | 1.33–2.43 | <0.001*** | 1.66 | 1.24–2.22 | <0.001*** |
| Marital status | | | | | | | | | |
| Living together | (Ref.) | | | (Ref.) | | | (Ref.) | | |
| Not living together | 0.71 | 0.44–1.16 | 0.175 | 1.23 | 0.87–1.74 | 0.233 | 1.00 | 0.72–1.37 | 0.98 |
| Enrolment in health insurance | | | | | | | | | |
| Not enrolled | (Ref.) | | | (Ref.) | | | (Ref.) | | |
| Enrolled | 3.77 | 1.75–8.14 | <0.001*** | 1.18 | 0.59–2.38 | 0.634 | 1.17 | 0.60–2.29 | 0.644 |
| Household size [person/household] | 0.96 | 0.82–1.13 | 0.639 | 1.01 | 0.91–1.13 | 0.823 | 1.02 | 0.92–1.14 | 0.708 |
| Household's wealth group | | | | | | | | | |
| Poor [b] | (Ref.) | | | (Ref.) | | | (Ref.) | | |
| Middle [c] | 2.40 | 1.21–4.77 | 0.012** | 1.60 | 1.07–2.39 | 0.021** | 1.72 | 1.18–2.52 | 0.005*** |
| Rich [d] | 1.90 | 1.08–3.31 | 0.025** | 1.37 | 0.96–1.94 | 0.083* | 1.70 | 1.21–2.40 | 0.002*** |

* $P < 0.10$

** $P < 0.05$

*** $P < 0.01$

[a] The interaction term of group and time pre-intervention (baseline) vs post-intervention (follow-up)

[b] Lower tertile of index (i.e., bottom 33%)

[c] Middle tertile of wealth index (i.e., middle 33%)

[d] Higher tertile of wealth index (i.e., top 33%)

Moreover, it should be carefully noted that the intervention package had been originally designed to priority target women from poor households. For the above two reasons, wealth-group-specific DID was examined, through estimating crude DID by wealth group of mothers' households (Fig 4). Again, ANC for the poor group was the only outcome variable that produced significant crude DID (11.4 percentage points; $P = 0.085$). For the other two outcome variables, no significant crude DID values were produced in all three economic level groups. Considering not only the importance of mothers from poor households as the Program's priority target populations but also a need for more precisely DID estimates, adjusted DID estimates among mothers from poor households were further calculated for the three key outcome variables.

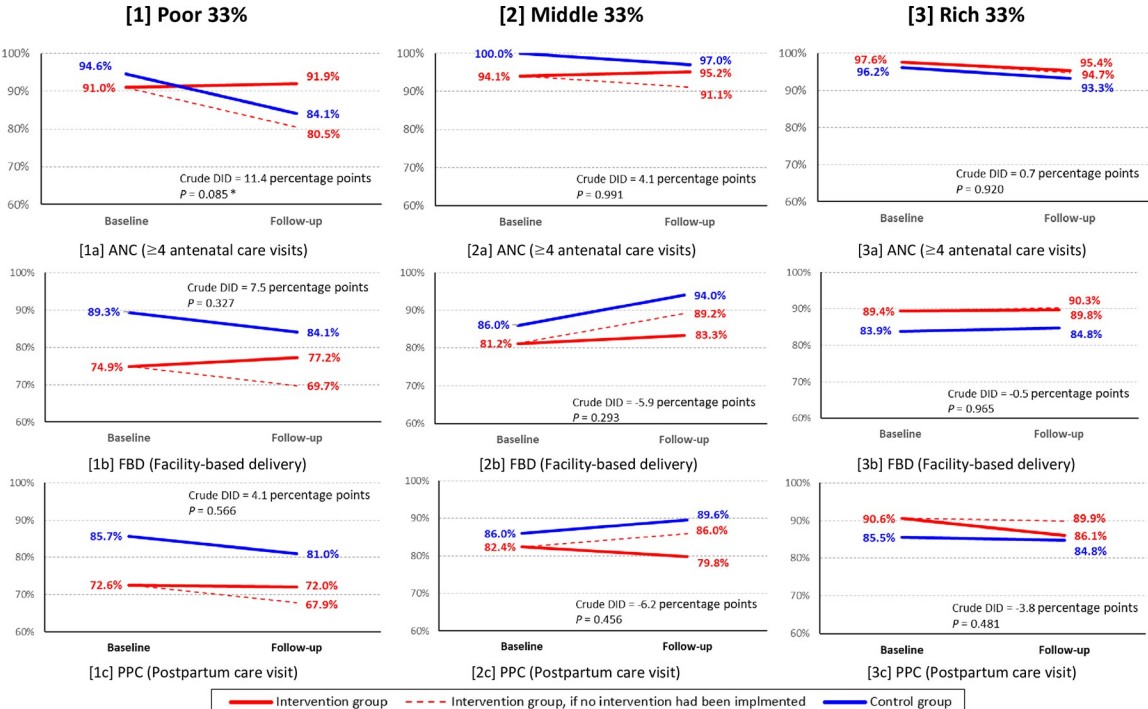

**Fig 4. Wealth-group-specific crude difference-in-differences for three outcome variables.** Crude DID is expressed in form of percentage points. * $P < 0.10$, ** $P < 0.05$, *** $P < 0.01$.

Table 5 shows the results of mixed-effect logistic regressions for the three outcome variables with an interaction between DID estimates and wealth groups (i.e., poor and rich). Mothers from poor households in the intervention group exhibited an increase in the likelihood of utilizing ANC services, although it was not significant (aOR = 3.63; $P = 0.231$). Also, our analysis did not reveal a variation in the effectiveness of the intervention package for the other outcomes across wealth tertiles.

## Discussion

Of the three types of essential maternal health service utilization (ANC, FBD and PPC), the intervention package improved only ANC visit coverage. A positive impact of the intervention package on FBD and PPC was not confirmed. Moreover, an adjusted DID estimate for ANC among mothers from poor households shows the positive trend, while it was not significant (aOR = 3.63; $P = 0.231$).

### Effectiveness in ANC coverage

Overall, the intervention package of the Program was effective only in increasing ANC coverage. The differences in DID estimates among the poor group did not produce statistically significant effect size, as indicated by the adjusted odds ratios (Table 5). These findings may suggest that a larger sample size would have uncovered a discernible trend of the impact of the intervention package across different wealth groups. Several earlier studies in low- and middle-income countries reported that health education and promotion contributed to an increase in the proportion of those having completed at least four ANC visits among rural pregnant women [13, 40, 41]. Generally, when health education and promotion activities were

**Table 5. Results of mixed-effect logistic regression for three outcome variables: Inter-wealth-group comparison analysis.**

| Independent variable | Model 1 | | | Model 2 | | | Model 3 | | |
|---|---|---|---|---|---|---|---|---|---|
| | ANC: ≥4 Antenatal care visits | | | FBD: Facility-based delivery | | | PPC: Postpartum care visit | | |
| | aOR | 95% CI | P-value | aOR | 95% CI | P-value | aOR | 95% CI | P-value |
| Intercept | 5.44 | 1.33–22.34 | 0.019** | 2.18 | 0.73–6.56 | 0.165 | 3.06 | 1.07–8.74 | 0.036** |
| **Group** | | | | | | | | | |
| Control | (Ref.) | | | (Ref.) | | | (Ref.) | | |
| Intervention | 0.95 | 0.28–3.26 | 0.932 | 1.11 | 0.47–2.61 | 0.816 | 1.28 | 0.58–2.83 | 0.545 |
| **Time** | | | | | | | | | |
| Baseline | (Ref.) | | | (Ref.) | | | (Ref.) | | |
| Follow-up | 0.67 | 0.25–1.76 | 0.413 | 1.21 | 0.70–2.09 | 0.490 | 1.01 | 0.58–1.73 | 0.984 |
| **Household's wealth group** | | | | | | | | | |
| Rich [a] | (Ref.) | | | (Ref.) | | | (Ref.) | | |
| Poor [b] | 0.95 | 0.24–3.82 | 0.946 | 1.49 | 0.68–3.24 | 0.320 | 1.00 | 0.48–2.09 | 0.997 |
| **Interactions** | | | | | | | | | |
| Interaction between group and wealth group | 0.50 | 0.09–2.80 | 0.433 | 0.34 | 0.13–0.92 | 0.034** | 0.40 | 0.15–1.05 | 0.064* |
| Interaction between time and wealth group | 0.42 | 0.08–2.15 | 0.300 | 0.98 | 0.34–2.83 | 0.966 | 1.11 | 0.41–3.02 | 0.832 |
| Difference in DID estimates among the rich | 1.12 | 0.24–5.17 | 0.884 | 1.38 | 0.53–3.56 | 0.509 | 0.97 | 0.39–2.41 | 0.944 |
| Difference in DID estimates among the poor (DDD) | 3.63 | 0.44–29.98 | 0.231 | 0.63 | 0.16–2.51 | 0.516 | 0.84 | 0.23–3.07 | 0.793 |
| **Maternal age [year]** | | | | | | | | | |
| < 20 | (Ref.) | | | (Ref.) | | | (Ref.) | | |
| 20–29 | 1.47 | 0.70–3.10 | 0.308 | 1.50 | 0.84–2.68 | 0.172 | 1.05 | 0.59–1.85 | 0.879 |
| 30–39 | 2.04 | 0.92–4.52 | 0.080* | 1.77 | 0.97–3.23 | 0.062* | 1.42 | 0.78–2.57 | 0.248 |
| ≥ 40 | 1.51 | 0.57–4.00 | 0.409 | 1.43 | 0.71–2.88 | 0.321 | 1.09 | 0.54–2.18 | 0.806 |
| **Education attainment** | | | | | | | | | |
| None / pre-primary education and primary education | (Ref.) | | | (Ref.) | | | (Ref.) | | |
| Secondary and higher education | 1.22 | 0.76–1.96 | 0.420 | 1.79 | 1.32–2.42 | <0.001*** | 1.66 | 1.24–2.22 | <0.001*** |
| **Marital status** | | | | | | | | | |
| Living together | (Ref.) | | | (Ref.) | | | (Ref.) | | |
| Not living together | 0.67 | 0.41–1.11 | 0.118 | 1.27 | 0.90–1.80 | 0.174 | 1.01 | 0.73–1.39 | 0.952 |
| **Enrolment in health insurance** | | | | | | | | | |
| Not enrolled | (Ref.) | | | (Ref.) | | | (Ref.) | | |
| Enrolled | 3.79 | 1.76–8.17 | <0.001*** | 1.11 | 0.55–2.24 | 0.766 | 1.12 | 0.57–2.2 | 0.744 |
| Household size [person/household] | 0.96 | 0.82–1.13 | 0.638 | 1.01 | 0.90–1.13 | 0.896 | 1.02 | 0.91–1.13 | 0.773 |

* P < 0.10

** P < 0.05

*** P < 0.01

[a] Higher than median of wealth index

[b] Lower than the median of wealth index

conducted by peers rather than by health workers [42], parents [43], or teachers [44, 45] and internet [46], its effectiveness became more conspicuous.

A study in Ethiopia indicated that peer education was one of the factors significantly associated with ANC coverage [28]. Another study in Zambia reported that a combination of ANC service availability and peer education significantly increased ANC coverage, compared with ANC service availability alone [20]. Thus, the findings of our study support the results of these earlier studies. Higher education and greater exposure to media are the facilitators for service utilization [47–52]. However, mothers from poor households are generally likely not only to be poorly educated and less literate [53, 54], but also to have limited access to media and

mobile phones [55, 56]. In fact, our study found the proportion of mothers having completed secondary education or higher was significantly smaller among those from poor households than among the others (*P* < 0.001). Thus, peer educators likely to be friendlier to mothers from poor households should be appropriate facilitators in more effectively encouraging them to make at least four ANC visits.

Probably, improvement of the quality of services through increasing clinical staff's skills alone has limited effectiveness in increasing ANC coverage. In rural areas where access to health facilities is often more challenging, the creation of service demands is an equally necessary intervention for increasing ANC4 coverage [11, 12, 57]. Note that the Program attempted to creating ANC service demands through peer education and promotion, while other previous projects did it through offering an incentive [11, 12, 58].

A significant increase in ANC coverage has been reported often among women from poor households [26, 59–61]. One of the possible reasons for this is that there is relatively greater room for an increase in ANC coverage among women from poor households [62], compared with those from middle-wealth or richer households. ANC coverages among middle-wealth and rich households are generally already high enough [63]. The results of our study are in the similar trend, by identifying 91.1%, 94.0%, and 97.6% of ANC coverages among those from poor households, middle-wealth households, and rich households, respectively.

## Effectiveness in FBD and PPC

This study did not identify the effectiveness of the intervention package in increasing service coverages of FBD and PPC not only among mothers in general but also even specifically among those from poor households.

The effectiveness of intervention projects aimed at increasing the coverage of ANC, FBD, and PPC differs according to the type, combination, and magnitude of interventions, and is mixed. Some interventions such as mass media campaigns and health education reported to significantly increase the service coverage of ANC but not of FBD [64, 65]. The intensive quality improvement initiative jointly implemented by District Health Management Team and respective health facilities improved service coverage of both ANC and FBD in Kenya [66]. Health promoters' intervention targeting the poorest population was successful in increasing service coverages of ANC, FBD and PPC [26, 58]. One study in Uganda found that an intervention package composed of community mobilization and capacity building of health workers increased service coverage of FBD but not of ANC [67], In Ethiopia, an intervention package composed of training of health workers and extension workers, rehabilitations of health centers, consumable supplies and community sensitization resulted in increased service coverage only of FBD [10].

It is reasonable that increasing ANC visits is less challenging and earlier realizable than increasing utilization of FBD and PPC services. This is because a majority of those having utilized FBD services must have made ANC visits while very few mothers not having made any ANC visits choose FBD. In other words, ANC attendance is one of the determinants of utilization of FBD services [68–76]. For the same logic, ANC and FBD service utilization are the primary and secondary determinants of utilization of PPC services, respectively [71, 77–79]. A systematic review of the studies in Ethiopia reported that mothers having attended one or more ANC were four times more likely to deliver at facilities and, then, receive PPC [80]. Moreover, deliveries are unpredictable events that can take place on rainy days, at night, and on weekends. Unpredictability of labor is a significant barrier to increasing FBD coverage [78].

More intensive and comprehensive intervention may increase FBD and PPC coverage. Improving quality of care is indispensable to increasing FBD coverage [74, 81, 82]. The

Program provided facility-based clinical health workers with the training on client-centered care that addresses four of the eight standards of quality of maternal care: (i) effective communication; (ii) respect and preservation of dignity; (iii) emotional support; and (iv) competent and motivated human resources [83]. However, training on other maternal care standards (e.g., evidence-based routine care practices and essential physical resource availability) might be additionally necessary to increase FBD coverage [10, 67]. Client-centered care, the second pillar of the intervention package, might not have been effective enough to adequately attract pregnant women to FBD and PPC services. This is because there are several other key barriers to FBD and PPC services utilization in Ghana, as reported earlier (i.e., greater informal payments to health workers, and inadequate health infrastructures, medical equipment and essential medicine) [84].

## Limitations of the study

There are four types of limitations of the study. First, there is possible sampling bias in both intervention and control groups. Since the lists of children were not readily available in the local civil registration systems, EPI registers were used as the sampling frame for both groups. This is because EPI registers were expected to cover almost all the children whose mothers were the study participants as 96.4% of children were vaccinated at least once through EPI in Eastern Region.

Second, the quality of baseline data could be less accurate because they were collected at the time of follow-up data collection, too. Though baseline data should be collected from both groups prior to the implementation of interventions generally in any DID studies, this study collected both baseline and follow-up data cross-sectionally only after the interventions was implemented. For this reason, the reliability of the baseline data particularly those derived from the respondents' memories (e.g., the number of ANC visits) should be less accurate.

Third, the COVID-19 pandemic made health services available in an intermittent manner during its pandemic period. This extraordinary situation sometimes created the cases where pregnant women and mothers returned home without receiving ANC, FBD, and/or PPC services at health facilities in Ghana [85–87]. They were caused largely by temporary reassignments of health workers from obstetric and gynecological department to other clinical departments at health facilities [85]. Moreover, some women must have been hesitant and reluctant in utilizing maternal care services for fear of possible infections with COVID-19 at health facilities. This might have led to underestimation of the effectiveness of the intervention package of the Program. If it had not been for the COVID-19 pandemic, the effectiveness of the intervention package would have been more precisely estimated. Thus, the COVID-19 pandemic undermined the generalizability of the results of the study.

Fourth, there are little possibilities of contamination of a series of interventions from the intervention group into the control group. This is because: (i) Atiwa West District (control group) and Suhum Municipality (intervention group) are not mutually adjacent; and (ii) people's movement was regulated and voluntarily refrained from during the COVID-19 pandemic. Yet, a certain level of the possibilities of contamination cannot be negated.

## Conclusion

Of the three outcome variables set, only the proportion of women having utilized at least four ANC services significantly increased in the intervention group, compared with the control group. We recommend the intervention package be scaled up either nationwide or selectively to the areas where greater room for improvement in ANC coverage is identified. Ghana has been in transition process of shifting the minimum number ANC visits from four to eight.

This indicates the additional need for an overall nationwide increase in the number of ANC visits. Thus, scaling up of the intervention package tested in this study is expected to help the transition be smoothly completed though ensuring increasing in the number of ANC visits.

## Acknowledgments

The authors gratefully thank Magnus Ebo Duncan and his team for their technical supports in data collection. This work is sincerely dedicated to all the reproductive-aged women in rural areas of Ghana.

## Author Contributions

**Conceptualization:** Hirotsugu Aiga, Yoshito Kawakatsu, Nobuhiro Kadoi, Kazuki Fujishima, Etsuko Yamaguchi.

**Data curation:** Hirotsugu Aiga, Nobuhiro Kadoi, Emmanuel Obeng, Frank Tabi Addai, Frederick Ofosu, Etsuko Yamaguchi.

**Formal analysis:** Hirotsugu Aiga, Yoshito Kawakatsu.

**Funding acquisition:** Kazuki Fujishima.

**Methodology:** Hirotsugu Aiga, Yoshito Kawakatsu, Etsuko Yamaguchi.

**Project administration:** Etsuko Yamaguchi.

**Resources:** Etsuko Yamaguchi.

**Supervision:** Hirotsugu Aiga, Etsuko Yamaguchi.

**Validation:** Mayumi Omachi.

**Writing – original draft:** Hirotsugu Aiga, Mayumi Omachi.

**Writing – review & editing:** Hirotsugu Aiga, Nobuhiro Kadoi, Emmanuel Obeng, Frank Tabi Addai, Frederick Ofosu, Kazuki Fujishima, Mayumi Omachi, Etsuko Yamaguchi.

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
