## [Decision Letter · Decision Letter 0]

14 Jul 2023

PONE-D-23-10669Effectiveness of community-based intervention package in maternal health service utilizations: A cross-sectional quasi-experimental study in rural GhanaPLOS ONE

Dear Dr. Aiga,

Thank you for submitting your manuscript to PLOS ONE. After careful consideration, we feel that it has merit but does not fully meet PLOS ONE’s publication criteria as it currently stands. Therefore, we invite you to submit a revised version of the manuscript that addresses the points raised during the review process.

Please find my comments and suggestions below. 

We look forward to receiving your revised manuscript.

Kind regards,

Mohammed Moinuddin, PhD

Academic Editor

PLOS ONE

Journal Requirements:

“The authors also thank Takeda Global CSR Program of Takeda Pharmaceutical Company Limited, for its financial support to the study.”

“This work was financially supported by Takeda Global CSR Program, Takeda Pharmaceutical Company Limited, JAPAN.”

“This work was financially supported by Takeda Global CSR Program, Takeda Pharmaceutical Company Limited, JAPAN.”

“EO, FTA, KF and EY were partially engaged either in planning or in implementing the Program. All the other authors (HA, YK, NK and MO) declare that they have no competing interests.

[Authors’ acronyms] Hirotsugu Aiga (HA); Yoshito Kawakatsu (YK); Nobuhiro Kadoi (NK); Emmanuel Obeng (EO); Frank Tabi Addai (FTA); Frederick Ofosu (FO);

Kazuki Fujishima (KF); Mayumi Omachi (MO); and Etsuko Yamaguchi (EY).”

Additional Editor Comments:

The financial disclosure does not include the required information. Please provide the full information required by the journal as below.

Funded studies

Enter a statement with the following details:

• Initials of the authors who received each award

• Grant numbers awarded to each author

• The full name of each funder

• URL of each funder website

• Did the sponsors or funders play any role in the study design, data collection and analysis, decision to publish, or preparation of the manuscript?

o NO - Include this sentence at the end of your statement: The funders had no role in study design, data collection and analysis, decision to publish, or preparation of the manuscript.

o YES - Specify the role(s) played.

How the consent was obtained is not clear in the ethical statement. This is a requirement. In the data availability statement, the authors said “risk of inappropriate use” which is a very grey terminology. Please be specific what does it mean and how the data can be used inappropriately. PLOS only requires submitting the anonymised data in a safe repository which is not publicly available for anyone to use. It is beneficial for other researchers to use anonymised data with a specific objective and through proper channel.

Reviewers' comments:

Reviewer's Responses to Questions

**Comments to the Author**

1. Is the manuscript technically sound, and do the data support the conclusions?

Reviewer #1: Partly

Reviewer #2: Partly

2. Has the statistical analysis been performed appropriately and rigorously? 

Reviewer #1: No

Reviewer #2: I Don't Know

3. Have the authors made all data underlying the findings in their manuscript fully available?

Reviewer #1: No

Reviewer #2: No

4. Is the manuscript presented in an intelligible fashion and written in standard English?

Reviewer #1: No

Reviewer #2: No

5. Review Comments to the Author

Reviewer #1: I am thankful for this opportunity of reading a very interesting research. I must emphasize that the manuscript will benefit immensely from a thorough editing to eliminate language-related and grammatical mistakes.

Abstract

Background

The main objective is not clearly written.

I would request the authors to consider writing something like this: “We examined the effectiveness of a community-based intervention package for pregnant women in increasing utilization of maternal health services. The package consisted of: (i)..., (ii)..., (iii)... and (iv) ...

Methods

I recommend the authors clarify how both baseline and follow-up data were collected simultaneously “after the implementation of the intervention package” (lines 102–104). How could baseline data be collected “after the implementation of the intervention package”? I found out the answer in lines 228–229, but readers should not be kept wondering about this. Thus, it needs to be clarified in the Abstract that by the word “simultaneously” the authors meant simultaneously in intervention and control sites.

The authors need to specify when the intervention was implemented (from X month to Y month of the year Z) in this section of the Abstract.

The authors need to mention the P-value that they considered for the level of statistical significance: P < 0.05 or a different one? I know that they have provided two reasons in lines 345–350, but here in the Abstract they need to mention that they considered P < 0.10 to indicate statistically significant associations.

Results

This section of the Abstract must be re-written. The authors need to present the answer to their main research question; i.e., the adjusted estimates for the three outcome variables.

I also recommend deleting lines 112–115 for the following two reasons:

i. whether the effectiveness of the intervention varied by categories of household wealth (assuming this to be a proxy for socio-economic status) was not the primary research question. In simpler words, whether household wealth or socio-economic status modifies the “effect” of the intervention on the outcome was not the primary research question as per the explanations provided by the authors in lines 439–445;

ii. the authors kept comparing the stratum-specific odds ratio for women from the “poorer” households with the overall odds ratio for wealth level. Instead, the stratum specific odds ratio for women from the “poorer” households should be compared against stratum-specific odds ratios for the other two wealth categories; i.e., odds ratios for women from the “middle-income” and “richer” households. Moreover, it was not clear whether the authors tested the interaction term incorporating “group” and “wealth level” in the mixed-effects logistic regression model (the legend below Table 3, line 433 does not indicate so). Without the interaction term “group × level” being tested and its P-value being statistically significant, it would be erroneous to infer that household wealth or socio-economic status modifies the “effect” of the intervention on the outcome.

Conclusion

Lines 118–120 read, “Peer health education activities and training existing health workers are likely to have synchronously contributed to an increase in antenatal care coverage. This is largely because peer educators are friendlier to the women from poor households”. I strongly recommend the authors consider deleting these two sentences as they have done no analysis to support these inferences. These inferences regarding how or through which mechanism(s) did the intervention work were not examined by the authors in the study. Besides, I wonder how the authors would know: i. whether “peer health education” and training of “existing health workers” acted synchronously, and ii. peer educators were “friendlier to the women from poor households”?

I would recommend the authors reiterate the main result and comment on the public health relevance of the extent of increase in maternal health care utilization brought on by the intervention in the Conclusion. It would be insightful if the authors comment on whether they think the intervention should be scaled up based on their findings. I would assume the overall impact to be smaller than anticipated after an intervention that ran for more than 16 months (as per lines 238–243).

Main text

Background

Lines 158–161, please provide the reference for these interventions being effective at the end of the sentence.

Lines 161–165, please re-frame this long sentence containing 45 words.

Line 167, please omit the word “physical”.

The authors argued that home visits carried out by “formal health workers” do not lead to “expected effectiveness” as “formal health workers” are “not necessarily interactive and friendly enough to local women” (lines 174–177). I briefly looked into the paper that they cites as a reference for this argument (reference number 11, a study from Ghana by Konlan et al.), and could not find the part of the paper that supports the authors’ arguments. I would request the authors to help me find the part of the paper on which the authors based their argument. Otherwise, the authors need to find a different reference to support their argument.

Line 192, please omit the word “in”.

Lines 200–203 read “The Program also strengthened the functions and capacity of the existing community health management committees, to enable them to continuously support both CHVs and facility-based health workers”. The authors need to briefly mention how did the Program strengthen the “functions and capacity” of the committees.

Customarily, a reader would expect to find out the specific objective(s) of the study in the last paragraph of the Background. However, this was lacking in the manuscript. I would recommend the authors do this instead of what they wrote in lines 213–215.

Methods

I would strongly recommend the authors shed light on any theoretical framework or model and empirical evidence from the same or similar settings that they used to inform and design the intervention. Otherwise, the approach would appear to be “top down” and shaped by the funder’s agenda, not by the local community’s needs.

Under the subsection “Interventions and outcome variables”, the authors need to provide information regarding the educational status of the Maternal and Child Health Promoters and Peer Educators.

In lines 302–305, the authors need to describe how the “capacity of community health management committees was strengthened” for carrying out “evidence-based planning, implementation, monitoring and evaluation”. The authors avoided describing what was actually done “under the third pillar”. This leaves readers open to speculation.

Line 354, please provide a reference that outlines how principal component analysis is used for computing wealth index or asset scores.

In lines 356–358, the authors wrote that “Then, all the households were categorized into five groups (i.e. wealth quintiles) according to the values of wealth index”. However, in the Results section, in Figure 3 captioned “Economic-group-specific crude difference-in-difference for three outcome variables”, they appear to have collapsed the bottom two quintiles into one single category (labelled “Poorer”) and the top two quintiles into another single category (labelled “Richer”). This is a serious discrepancy in my opinion. I strongly recommend the authors either: i. present the DID for all five quintiles, or ii. they convert the values of wealth index into tertiles instead of quintiles and present the DID for all three tertiles.

It would be transparent, if the authors clearly mention which variables were considered random-effect and which were considered fixed-effect in the mixed-effects logistic regression.

I also suggest the authors mention how they assessed the adequacy or fit of the logistic regression models and how they checked for collinearity.

The authors wrote that they had access to attributes that could identify individual participants during and after data collection (lines 368–369). The authors need to mention under Ethical considerations how they have ensured the confidentiality of identifying attributes and personal information as well as what would happen to these in future. The authors need to mention whether participation was voluntary and whether the participants retained their right to withdraw throughout the study.

Results

It is a common practice in writing the Results section that only the findings are presented not any interpretation(s) of the finding(s). Nevertheless, the authors resorted to presenting interpretations and inferences in the Results section (lines 396–400 and 401–405). The authors need to remove these.

As mentioned above, the authors either: i. present the crude DID for all five quintiles, or ii. they convert the values of wealth index into tertiles instead of quintiles and present the DID for all three tertiles in Figure 3. Similarly, Table 4 should demonstrate the estimates for all five or three wealth groups, not just the estimates among “poorer” women. I also think both in Tables 3 and 4, the estimates for “Maternal age”, “Education attainment”, “Marital status”, “Enrolment in health insurance” and “Household size” should not be presented as the main research question does not deal with these variables. Not presenting these estimates would also make the tables less busy.

Discussion

In the first paragraph of this section, the authors need to interpret the difference in difference estimator in a way that would give the readers an impression about how much improvement was brought on by the intervention and what this means from an “effect size” perspective. Please see the comment for the Conclusion

Lines 473–475 read, “This indicates that ANC coverage significantly increased between pre- and post-intervention stages particularly among women from poor households in the intervention group”. I do not agree with how the sentence has been framed. The authors should consider writing something like this: out of the three aspects of maternal health service utilization, the intervention improved only the ANC visit coverage ... A positive impact of the intervention on ... ... ... was not observed.

Furthermore, I think the word “significantly” in line 473 is ambiguous: did the authors mean statistical significance? If so, they should write it clearly. Or, did the authors use “significantly” to indicate a considerably large “effect size”? In that case, the authors need to elaborate further.

I strongly recommend the authors remove what they wrote in lines 565–601. While it is very interesting to explore what might have happened when there were health systems disruptions from COVID-19, this should be reserved for another study. Moreover, the authors used the word “resilience”, and if I am to assume this to be a construct, then the question arises which variable captured this construct. I could not find any variable that systematically and validly captured resilience. In fact, the authors performed no specific analysis to support what they discussed in line 565–601; which makes it akin to speculation. The authors argued that “People’s overall hesitancy to go out should have made pregnant women in the control group refrain from making antenatal care visits. Despite generally increased hesitancy to go out, the proportion of those having made at least four ANC visits slightly increased in the intervention group” (lines 585–589). There could be other confounding factors behind this small increase in proportions (unadjusted, bivariate estimate) of those completing at least four ANC visits among “poorer” women that were not possible to capture because of the study design.

I would prefer a Conclusion section where the authors present the principal finding in plain English and comment on the public health relevance of the extent of increase in maternal health care utilization brought on by the intervention and its scalability.

Reviewer #2: The study is on an important subject with precious information on interventions and women's health.

Regarding the abstract, it could be revised to be more on point and send a clearer message. Also the background and methods need, I would recommend to rewrite those sections and some suggestions are in the document, which can be found as an attachment. Additionally as it is based on an intervention, I would recommend to make a graphical depiction on how the intervention was carried out.

This study might be read, and used also in other contexts and thus it is important to include context specific information on current healthcare services for women of reproductive age.

6. PLOS authors have the option to publish the peer review history of their article (what does this mean?). If published, this will include your full peer review and any attached files.

Reviewer #1: No

Reviewer #2: No

---

## [Author Response · Author response to Decision Letter 0]

22 Dec 2023

RESPONSES TO THE REVIEWERS’ COMMENTS

We are extremely grateful to the editor and reviewers for their comments on the manuscript. We have significantly revised the manuscript based on those useful comments. Please refer to the following point-by-point-based responses to the respected comments, along with revised manuscript. Please note that the changes in the revised manuscript (i.e. main text, tables/figures, and reference list) have been shown in: (i) red in this file for “Responses to Reviewers” as follows; (ii) track changes in the file “Revised Manuscript with Track Changes”; and (iii) red in the cleaned revised file “Revised Manuscript”.

COMMENTS FROM REVIEWER #1

[Comment 1-1] I am thankful for this opportunity of reading a very interesting research. I must emphasize that the manuscript will benefit immensely from a thorough editing to eliminate language-related and grammatical mistakes.

[Response 1-1] We appreciate Reviewer #1 for his/her overall comments on this paper. We have addressed the specific comments and suggestions made by Reviewer #1 as follows.

[Comment 1-2] Abstract/Background: The main objective is not clearly written.

I would request the authors to consider writing something like this: “We examined the effectiveness of a community-based intervention package for pregnant women in increasing utilization of maternal health services. The package consisted of: (i)..., (ii)..., (iii)... and (iv) ...

[Response 1-2] Thanks for this concrete suggestion. The first sentence in ‘Background’ of Abstract has been revised according to this suggestion. Now, the manuscript reads “We examined the effectiveness of a community-based intervention package that targeted pregnant women for increasing utilizations of maternal health services. The intervention package was…..” (see lines 93-95 “Revised Manuscript with Track Changes”; lines 93-95 of “Revised Manuscript”).

[Comment 1-3] Abstract/Methods: I recommend the authors clarify how both baseline and follow-up data were collected simultaneously “after the implementation of the intervention package” (lines 102–104). How could baseline data be collected “after the implementation of the intervention package”? I found out the answer in lines 228–229, but readers should not be kept wondering about this. Thus, it needs to be clarified in the Abstract that by the word “simultaneously” the authors meant simultaneously in intervention and control sites.

[Response 1-3] This comment from Reviewer #1 is reasonable. The necessary revision has been made accordingly. Now, the manuscript reads “A cross-sectional household survey was conducted in May 2022. We sampled four groups of women: (i) intervention at baseline; (ii) intervention at follow-up; (iii) control at baseline; and (iv) control at follow-up”. (see lines 101-103 of “Revised Manuscript with Track Changes; lines 101-103 of “Revised Manuscript”). Please note that the word count limit for Abstract set by PLoS One (=300 words) does not allow us to write more details on how baseline and follow-up data were collected in a cross-sectional manner. 

[Comment 1-4] Abstract/Methods: The authors need to specify when the intervention was implemented (from X month to Y month of the year Z) in this section of the Abstract.

[Response 1-4] To respond to this suggestion from Reviewer #1, an additional sentence “The intervention package was implemented in Suhum Municipality, Ghana, from March 2019 to April 2022.” has been inserted in the Background of Abstract (see lines 94-96 of “Revised Manuscript with Track Changes”; lines 94-96 of “Revised Manuscript”).

[Comment 1-5] Abstract/Methods: The authors need to mention the P-value that they considered for the level of statistical significance: P < 0.05 or a different one? I know that they have provided two reasons in lines 345–350, but here in the Abstract they need to mention that they considered P < 0.10 to indicate statistically significant associations.

[Response 1-5] Thanks for this careful comment. Based on the comment, a new sentence has been newly inserted in Methods of Abstract, accordingly, i.e. “Significance levels were set at the values of 0.10, 0.05, and 0.01, since the aforementioned service coverages had been already too high to have room for an increase at the significance level of 0.05.” (see lines 108-133 of “Revised Manuscript with Track Changes”; lines 108-110 of “Revised Manuscript”).

[Comment 1-6] Abstract/Results: This section of the Abstract must be re-written. The authors need to present the answer to their main research question; i.e., the adjusted estimates for the three outcome variables. I also recommend deleting lines 112–115 for the following two reasons: (i) whether the effectiveness of the intervention varied by categories of household wealth (assuming this to be a proxy for socio-economic status) was not the primary research question. In simpler words, whether household wealth or socio-economic status modifies the “effect” of the intervention on the outcome was not the primary research question as per the explanations provided by the authors in lines 439–445; (ii) the authors kept comparing the stratum-specific odds ratio for women from the “poorer” households with the overall odds ratio for wealth level. Instead, the stratum specific odds ratio for women from the “poorer” households should be compared against stratum-specific odds ratios for the other two wealth categories; i.e., odds ratios for women from the “middle-income” and “richer” households. 

[Response 1-6] Thanks for this constructive comment. Having agreed to the reviewer’s suggestion, lines 112-115 in the previous manuscript have been deleted. Then, ‘Results’ of Abstract has been rewritten. Now, the manuscript reads “The proportion of women having utilized at least four ANC services produced significant DID in both crude and adjusted estimates. The proportions of women having utilized facility-based delivery services and post-partum care services produced significant DID in neither crude nor adjusted estimates.” (see lines 136-139 of “Revised Manuscript with Track Changes”; lines 113-116 of “Revised Manuscript”).

[Comment 1-7] Abstract/Results: Moreover, it was not clear whether the authors tested the interaction term incorporating “group” and “wealth level” in the mixed-effects logistic regression model (the legend below Table 3, line 433 does not indicate so). Without the interaction term “group × level” being tested and its P-value being statistically significant, it would be erroneous to infer that household wealth or socio-economic status modifies the “effect” of the intervention on the outcome.

[Response 1-7] Thanks for the comment. Table 3 was created for the purpose of showing overall effectiveness of the intervention package among all the respondents regardless of household’s wealth levels. In other words, it is not intended to show the comparison of the effectiveness between the poorer, middle-income, and richer. Therefore, it is not necessary to add the interaction terms of ‘Group x Household’s wealth level’ in the mixed-effect logistic regression models for Table 3. Thus, the way of analyzing data for Table 3 does not change. Please note that the categorization of ‘Household’s wealth groups was corrected based on the reviewer’s reasonable comment in [Comment 1-20]. However, the reviewer’s suggestion of inclusion of the interaction term has been usefully applied to calculation of wealth-level-specific DID in Table 4.

[Comment 1-8] Abstract/Conclusion: Lines 118–120 read, “Peer health education activities and training existing health workers are likely to have synchronously contributed to an increase in antenatal care coverage. This is largely because peer educators are friendlier to the women from poor households”. I strongly recommend the authors consider deleting these two sentences as they have done no analysis to support these inferences. These inferences regarding how or through which mechanism(s) did the intervention work were not examined by the authors in the study. Besides, I wonder how the authors would know: i. whether “peer health education” and training of “existing health workers” acted synchronously, and ii. peer educators were “friendlier to the women from poor households”?

I would recommend the authors reiterate the main result and comment on the public health relevance of the extent of increase in maternal health care utilization brought on by the intervention in the Conclusion. It would be insightful if the authors comment on whether they think the intervention should be scaled up based on their findings. I would assume the overall impact to be smaller than anticipated after an intervention that ran for more than 16 months (as per lines 238–243).

[Response 1-8] Based on this strong recommendation, ‘Conclusions’ of Abstract has been totally rewritten. Implication of the nationwide scaling-up of the intervention package has been added. Now, the manuscript reads “Of the three outcome variables set, only the proportion of women having utilized at least four ANC services significantly increased in the intervention group, compared with the control group. Ghana has been in transition process of shifting the minimum number ANC visits from four to eight. Thus, nation-wide scaling up of the intervention package is expected to help the transition be smoothly completed though ensuring increasing in the number of ANC visits.” (see lines 141-146 of “Revised Manuscript with Track Changes”; lines 119-124 of “Revised Manuscript”). We agree to the reviewer’s view that the interruption of the intervention package due to COVID-19 pandemic made the overall impact actually smaller and less significant than anticipated. Yet, the word count limit for Abstract (=300 words) does not allow us to write on this point. Hope the reviewer will kindly understand this.

[Comment 1-9] Main text/Background: Lines 158–161, please provide the reference for these interventions being effective at the end of the sentence.

[Response 1-9] To respond to this comment, three references have been added to the sentence “To reduce a significantly greater number of maternal deaths in Sub-Saharan Africa, timely and quality antenatal care (ANC) and facility-based delivery (FBD), post-partum care (PPC), Emergency Obstetric Care, and family planning are imperative.” I.e. (i) Ref #5 - Sexual and Reproductive Health and Research, WHO. Maternal mortality: Evidence brief [Internet]. 2019. Available from: https://www.who.int/publications/i/item/WHO-RHR-19.20

(ii) Ref #6 - Paxton A, Maine D, Freedman L, Fry D, Lobis S. The evidence for emergency obstetric care. Intl J Gynecology & Obste. 2005 Feb;88(2):181–93.; and

(iii) Ref #7 - Levine R, Langer A, Birdsall N, Matheny G, Wright M, Bayer A. Contraception. In: Jamison DT, Breman JG, Measham AR, Alleyne G, Claeson M, Evans DB, et al., editors. Disease Control Priorities in Developing Countries [Internet]. 2nd ed. Washington (DC): The International Bank for Reconstruction and Development / The World Bank; 2006 [cited 2023 Oct 18]. Available from: http://www.ncbi.nlm.nih.gov/books/NBK11771/

(see line 213 and 907-918 of “Revised Manuscript with Track Changes”; line 164 and 672-683 of “Revised Manuscript”)

[Comment 1-10] Main text/Background: Lines 161–165, please re-frame this long sentence containing 45 words.

[Response 1-10] Thanks for the comment. The same/similar comment was raised by Reviewer #2 in [Comment 2-7]. Based on this reasonable comment from both reviewers, the sentence has been shortened, simplified and separated into two sentences. Accordingly, the number of words of the two sentences has been reduced from 45 to 32. Now, it reads “Poorer access to health facilities, inadequate quality of health services, and socio-economic/ cultural barriers are as the major obstacles in increasing maternal health service utilizations. This prevents reproductive-aged women from life-saving services” (see lines 213-216 of “Revised Manuscript with Track Changes”; lines 164-167 of “Revised Manuscript”). See the [Response 2-7], too.

[Comment 1-11] Main text/Background: Line 167, please omit the word “physical”.

[Response 1-11] Thanks for this suggestion. Having agreed to the reviewer’s suggestion, “physical” has been deleted. (see line 218 of “Revised Manuscript with Track Changes”; line 169 of “Revised Manuscript”).

[Comment 1-12] Main text/Background: The authors argued that home visits carried out by “formal health workers” do not lead to “expected effectiveness” as “formal health workers” are “not necessarily interactive and friendly enough to local women” (lines 174–177). I briefly looked into the paper that they cites as a reference for this argument (reference number 11, a study from Ghana by Konlan et al.), and could not find the part of the paper that supports the authors’ arguments. I would request the authors to help me find the part of the paper on which the authors based their argument. Otherwise, the authors need to find a different reference to support their argument.

[Response 1-12] Many thanks for carefully reviewing this part. We agree that the reference “Konlan et al. 2021” does nor clearly indicate poorer client-friendliness and interactivity of formal health workers’ demand creation (e.g. household visitis). The referencce “Konlan D et al. 2021” has been replaced by three new references, i.e. (i) Ref #15 – Moyer CA, Adongo PB, Aborigo RA, Hodgson A, Engmann CM. ‘They treat you like you are not a human being’: Maltreatment during labour and delivery in rural northern Ghana. Midwifery. 2014 Feb;30(2):262–8.; (ii) Ref #16 – Dapaah JM, Nachinaab JO. Sociocultural Determinants of the Utilization of Maternal Health Care Services in the Tallensi District in the Upper East Region of Ghana. Advances in Public Health. 2019 Feb 10;2019:1–11.; and (iii) Ref #17 – Amu H, Nyarko SH. Satisfaction with Maternal Healthcare Services in the Ketu South Municipality, Ghana: A Qualitative Case Study. BioMed Research International. 2019 Apr 10;2019:1–6.

(see line 243 and 953-963 of “Revised Manuscript with Track Changes”; line 179 and 713-723 of “Revised Manuscript”)

[Comment 1-13] Main text/Background: Line 192, please omit the word “in”.

[Response 1-13] Thanks for finding this typo. This “in” has been deleted, accordingly (see line 271 of “Revised Manuscript with Track Changes”; line 194 of “Revised Manuscript”).

[Comment 1-14] Main text/Background: Lines 200–203 read “The Program also strengthened the functions and capacity of the existing community health management committees, to enable them to continuously support both CHVs and facility-based health workers”. The authors need to briefly mention how did the Program strengthen the “functions and capacity” of the committees.

[Response 1-14] Thanks for this comment. An additional sentence has been newly inserted after this sentence “Capacity strengthening of community health management committees was conducted through a series of lectures, groups works, and role plays.” (see lines 282-284 of “Revised Manuscript with Track Changes”; lines 205-207 of “Revised Manuscript”).

[Comment 1-15] Main text/Background: Customarily, a reader would expect to find out the specific objective(s) of the study in the last paragraph of the Background. However, this was lacking in the manuscript. I would recommend the authors do this instead of what they wrote in lines 213–215.

[Response 1-15] We are thankful to the reviewer for this suggestion. An additional sentence “Specific objectives of the study are to estimate the degree of increases in proportions of women having utilized: (i) at least four ANC visits; (ii) FBD; and (iii) PNC.” has been inserted in the location the reviewer indicated (see lines 307-309 of “Revised Manuscript with Track Changes”; line 227-229 of “Revised Manuscript”).

[Comment 1-16] Methods: I would strongly recommend the authors shed light on any theoretical framework or model and empirical evidence from the same or similar settings that they used to inform and design the intervention. Otherwise, the approach would appear to be “top down” and shaped by the funder’s agenda, not by the local community’s needs.

[Response 1-16]

---

## [Decision Letter · Decision Letter 1]

26 Mar 2024

PONE-D-23-10669R1Effectiveness of a community-based intervention package in maternal health service utilizations: A cross-sectional quasi-experimental study in rural GhanaPLOS ONE

Dear Dr. Aiga,

Thank you for submitting your manuscript to PLOS ONE. After careful consideration, we feel that it has merit but does not fully meet PLOS ONE’s publication criteria as it currently stands. Therefore, we invite you to submit a revised version of the manuscript that addresses the points raised during the review process.

We look forward to receiving your revised manuscript.

Kind regards,

Mohammed Moinuddin, PhD

Academic Editor

PLOS ONE

Journal Requirements:

Reviewers' comments:

Reviewer's Responses to Questions

**Comments to the Author**

1. If the authors have adequately addressed your comments raised in a previous round of review and you feel that this manuscript is now acceptable for publication, you may indicate that here to bypass the “Comments to the Author” section, enter your conflict of interest statement in the “Confidential to Editor” section, and submit your "Accept" recommendation.

Reviewer #1: All comments have been addressed

Reviewer #3: All comments have been addressed

2. Is the manuscript technically sound, and do the data support the conclusions?

Reviewer #1: Yes

Reviewer #3: Partly

3. Has the statistical analysis been performed appropriately and rigorously? 

Reviewer #1: Yes

Reviewer #3: No

4. Have the authors made all data underlying the findings in their manuscript fully available?

Reviewer #1: (No Response)

Reviewer #3: Yes

5. Is the manuscript presented in an intelligible fashion and written in standard English?

Reviewer #1: Yes

Reviewer #3: Yes

6. Review Comments to the Author

Reviewer #1: I thank the authors for their substantial revision. My final, minor suggestions are as follows.

Line 94: please omit the "s" in the word "utilizations".

Line 101: please put a comma after the word "cross-sectional".

Line 109: please consider writing "had already been" instead of "had been already".

Line 113-116: please consider writing "The proportion of women completing at least four ANC visits displayed statistically significant DID in both crude and adjusted analyses. The proportions of women utilizing facility-based delivery services and post-partum care services did not display statistically significant DIDs.".

Line 123-124: please consider writing "... the transition be smooth by increasing the number of ANC visits.".

Reviewer #3: The authors made great efforts to overcome difficulties in conducting the impact evaluation during the pandemic and provide rigorous estimates. The reviewer was invited after the revised manuscript was submitted, i.e., did not participate in the review process at the initial submission. Therefore, the reviewer mainly commented on the points that other reviewers did not address in the first round of the peer review process.

1. Rationales of conducting this study: The authors might want to add explanations in the Background section regarding why this intervention was required in the study site where maternal health coverage was high. The authors added that “Poorer access to health facilities, inadequate quality of health services, and socio-economic/ cultural barriers are the major obstacles in increasing maternal health service utilizations. This prevents reproductive aged women from reaching life-saving services.” (Lines 164-) However, the authors might want to re-examine if this statement is valid to maternal health service provision situations in the study site. Particularly, they would need to explain that the intervention package benefited women who could not receive maternal health services while most women received these services.

2. Design effect: The authors set the design effect at 1.3 (Line 363) for the sample size calculation, seemingly without reference articles. As across-cluster differences may be large in an outcome of receiving health services, compared to a health outcome, this design effect might be too small. Readers may have an impression that a reason for insignificance in DID estimators at a 5% significance level might be because of insufficient sample size, not “the service coverages of the outcome variables had been already too high to have room for an increase 387 at the significance level of 0.05 (e.g., 93.1% of ANC in the intervention group at baseline).” (Lines 386-) The authors might want to justify the design effect level and add a limitation statement if the design effect was smaller than an adequate level. As the authors might have information regarding the high level of maternal health service coverage at the study site as the project funder, the high level of maternal health service coverage at baseline might not be a good reason for loosening the significance level to 10%. Rather than that, logistic or financial constraints in the impact evaluation might justify setting the significance level at 10%, particularly for behavioral studies.

3. Significance testing in socio-demographic and socio-economic characteristics: The authors explained that “Significant differences were unexpectedly detected in the proportions and mean values of 10 of 12 characteristic variables between the four groups (i.e., intervention group at baseline, intervention group at follow-up, control group at baseline, and control group at follow-up).” (Lines 441-) It is unclear why these four groups should have been compared. This approach seems to mix up detecting differences in baseline characteristics between the arms, detecting differences in follow-up characteristics between the arm, and detecting changes in characteristics between baseline and follow-up within the arm. If the authors needed to check the balance of the characteristics between the arms, they might want to present significance testing results at the baseline and at the follow-up separately.

4. Possible contamination: Were Suhum Municipality and Atiwa West District locations close with each other? Is it possible that people in Atiwa West District used maternal health services in Suhum Municipality or received part of the intervention package? The authors might want to expand explanations on the choice of the intervention and control group sites considering contamination possibilities.

5. Terminology: Is the word “difference-in-differences” more common than “difference-in-difference,” which the authors used? Please disregard this comment if the authors had a good reason to call their method difference-in-difference.

6. Estimation of Wealth-group-specific difference-in-differences: The approach used for the analysis of heterogeneity in DID across different wealth groups is so-called difference-in-difference-in-differences, or DDD (for example, Olden and Møen [2022], https://academic.oup.com/ectj/article/25/3/531/6545797). According to Table 5, the authors’ specification may include the variables of group, time, wealth, and DDD (group*time*wealth). Compared to Equation 3.1 in Olden and Møen (2022), the model in Table 5 might miss the interaction terms group*wealth and time*wealth (the interaction term of time*group may be captured as “Adjusted DID estimates among the poor”). Consequently, DDD estimator in Table 5 might cause a bias by capturing two interaction terms (group*wealth and time*wealth), in addition to the appropriate DDD (group*time*wealth). Please check the model in Table 5 to ensure if it was specified appropriately and possibly re-estimate DDD if needed.

7. Percent or percentage point change: In Figures 3 and 4, the authors presented crude DID as a percent form. However, it looks like it should have been “percentage points,” as this is the subtraction of percentages. Please find the following example for details: http://sumn.org/downloads/Percentage_Change.pdf.

7. PLOS authors have the option to publish the peer review history of their article (what does this mean?). If published, this will include your full peer review and any attached files.

Reviewer #1: No

Reviewer #3: No

---

## [Author Response · Author response to Decision Letter 1]

27 Apr 2024

RESPONSES TO THE REVIEWERS’ COMMENTS

We are extremely grateful to the editor and reviewers for their comments on the manuscript. We have revised the manuscript based on those useful comments. Please refer to the following point-by-point-based responses to the respected comments, along with revised manuscript. Please note that the changes in the revised manuscript (i.e. main text, tables/figures, and reference list) have been shown in: (i) red in this file for “Responses to Reviewers” as follows; (ii) track changes in the file “Revised Manuscript with Track Changes”; and (iii) red in the cleaned revised file “Revised Manuscript”.

COMMENTS FROM REVIEWER #1

[Comment 1-1] Line 94: please omit the "s" in the word "utilizations".

[Response 1-1] Thanks for this suggestion. This “s” has been deleted, accordingly.

Reviewer #1: I thank the authors for their substantial revision. My final, minor suggestions are as follows.

[Comment 1-2] Line 101: please put a comma after the word "cross-sectional".

[Response 1-2] Thanks. Yet, we have never seen the term “A cross-sectional household, survey” and we think it looks awkward. We found the term “A cross-sectional household survey” (without a comma after ”cross-sectional”) in a number of article titles. For instance “Seeking and reaching emergency care: A cross sectional household survey across two Liberian counties” (PLOS Global Public Health 3 (11): e0002629 [DOI: 10.1371/journal.pgph.0002629]). Thus, we have decided not to insert a comma.

[Comment 1-3] Line 109: please consider writing "had already been" instead of "had been already".

[Response 1-3] Thanks for this suggestion. The order of these two words have been switched, accordingly. 

[Comment 1-4] Line 113-116: please consider writing "The proportion of women completing at least four ANC visits displayed statistically significant DID in both crude and adjusted analyses. The proportions of women utilizing facility-based delivery services and post-partum care services did not display statistically significant DIDs.".

[Response 1-4] Thanks for this suggestion. These two sentences have been replaced by "The proportion of women completing at least four ANC visits displayed significant DID in both crude and adjusted analyses. The proportions of women utilizing facility-based delivery services and post-partum care services did not display significant DIDs.", accordingly. Note that the term “statistically” was not inserted despite the reviewer’s suggestion. This is because the term “significant” in the texts on statistical analyses in any public health papers automatically means “statistically significant”. As a common practice, not “statistically significant” but simply “significant” has been used by public health papers. Hope the reviewer will agree to this view.

[Comment 1-5] Line 123-124: please consider writing "... the transition be smooth by increasing the number of ANC visits."

[Response 1-5] Thanks for this suggestion. The phrase “... the transition be smoothly completed through ensuring the number of ANC visits” has been replaced by "... the transition be smooth by increasing the number of ANC visits ", accordingly.

COMMENTS FROM REVIEWER #3

[Comment 3-1] Rationales of conducting this study: The authors might want to add explanations in the Background section regarding why this intervention was required in the study site where maternal health coverage was high. The authors added that “Poorer access to health facilities, inadequate quality of health services, and socio-economic/ cultural barriers are the major obstacles in increasing maternal health service utilizations. This prevents reproductive aged women from reaching life-saving services.” (Lines 164-) However, the authors might want to re-examine if this statement is valid to maternal health service provision situations in the study site. Particularly, they would need to explain that the intervention package benefited women who could not receive maternal health services while most women received these services.

[Response 3-1] When the intervention package was designed and proposed in 2018-2020, the MCH service coverages in Eastern Region were overall low enough according to District Health Information System 2 (DHIMS2) (see (a) in table below). Thus, we assumed there was a critical need for the intervention package. Then, it was only after completion of data collection for this study when such high coverages of ANC, FBD and PPC as shown in Figure 3 in the manuscript were found (see also (b1) and (b2) in table below). Ghana Demographic and Health Survey 2022 whose data were collected after our study also reported equally high coverages for both Eastern Region and entire Ghana, too (see also (c1) and (c2) in table below). Though DHIMS data probably might have underestimated these coverages, they are the only available coverage data for Eastern Region we could rely on. There is no great gap between baseline data for our study (ie, (b1) and (b2)) and DHS2022 data (ie, (c1)) in Eastern Region. Similarly, there is no great gap in the coverages between Eastern Region (c1) and entire Ghana (c2) in DHS2022 data. This implies that there is definitely room and need for further improvement of these coverages in Eastern Region (incl. Suhum Municipality) as well as other regions of Ghana. Having considered the aforementioned complicated background on service utilization rates, we decided not to add any texts in the main text, to avoid unnecessarily presenting complicated local background information that may confuse the audience and rather maintain its simplicity. 

 (a) During intervention designing: DHIMS in Eastern Region, [2018]a (b) Our study: Baseline data of our survey in 2 districts in Eastern Region [5-29 May 2022] (c) After our study: Ghana DHS 2022 [17 Oct 2022 – 14 Jan 2023] b

 (b1) Intervention group (b2) Control group (c1) Eastern Region (c2) Ghana

+4ANC 55.8% 93.1% 96.7% 88.0% 87.8%

FBD 55.9% 79.4% 85.6% 89.5% 86.2%

PPC 62.4% 78.6% 85.6% 95.0% 87.0%

a [Source] District Health Information System 2 (DHIMS2)

b [Source] Ghana Statistical Service. Ghana demographic and health survey 2022. Accra: GSS; 2024.

[Comment 3-2] Design effect: The authors set the design effect at 1.3 (Line 363) for the sample size calculation, seemingly without reference articles. As across-cluster differences may be large in an outcome of receiving health services, compared to a health outcome, this design effect might be too small. Readers may have an impression that a reason for insignificance in DID estimators at a 5% significance level might be because of insufficient sample size, not “the service coverages of the outcome variables had been already too high to have room for an increase 387 at the significance level of 0.05 (e.g., 93.1% of ANC in the intervention group at baseline).” (Lines 386-) The authors might want to justify the design effect level and add a limitation statement if the design effect was smaller than an adequate level. As the authors might have information regarding the high level of maternal health service coverage at the study site as the project funder, the high level of maternal health service coverage at baseline might not be a good reason for loosening the significance level to 10%. Rather than that, logistic or financial constraints in the impact evaluation might justify setting the significance level at 10%, particularly for behavioral studies.

[Response 3-2] Thanks for the clarification. Despite the reviewer’s concern, we carefully set the design effect at 1.3 and further verified its validity, by referring to those reported in Ghana Demographic and Health Survey 2022. The DHS reported the values of design effect for the respective key variables specifically for Eastern Region, where we collected the data for this study. I.e., (i) 1.126 for ≥4ANC visits; (ii) 1.374 for facility-based delivery (FBD); and (iii) 1.056 for postpartum care (PPC). To ensure the adequacy of design effect for all the three outcome variables, the design effect was set at 1.3, by employing the greatest one of the three. Then, “Ghana Statistical Service. Ghana demographic and health survey 2022. Accra: GSS; 2024. https://www.dhsprogram.com/pubs/pdf/FR387/FR387.pdf (accessed April 9, 2024)” has been added as the reference for justifying the design effect of 1.3, accordingly. As the reviewer mentioned high maternal health service coverage at baseline might not be an appropriate reason for loosening the significance level to 10%. Yet, it might be, too. It is difficult to objectively judge whether high maternal health service coverage is an appropriate reason for loosening the significance level or not. This is because there is no standard method for doing so. Moreover, many DID studies use the three significance levels (i.e. 0.10, 0.05, and 0.01). For example, A El-Shal, P Cubi-Molla, M Jofre-Bonet Accreditation as a quality‑improving policy tool: family planning, maternal health, and child health in Egypt. The European Journal of Health Economics, 2021; 22 (1): 115–139.

[Comment 3-3] Significance testing in socio-demographic and socio-economic characteristics: The authors explained that “Significant differences were unexpectedly detected in the proportions and mean values of 10 of 12 characteristic variables between the four groups (i.e., intervention group at baseline, intervention group at follow-up, control group at baseline, and control group at follow-up).” (Lines 441-) It is unclear why these four groups should have been compared. This approach seems to mix up detecting differences in baseline characteristics between the arms, detecting differences in follow-up characteristics between the arm, and detecting changes in characteristics between baseline and follow-up within the arm. If the authors needed to check the balance of the characteristics between the arms, they might want to present significance testing results at the baseline and at the follow-up separately.

[Response 3-3] Thank you for your insightful comments. In response to your feedback, we have revised Table 3, to more clearly delineate the comparisons made between the groups. We now present the results of significance testing for socio-demographic and socio-economic characteristics separately for both baseline and follow-up. This revision allows for a clearer assessment of the comparability between the intervention and control groups at each time point. We appreciate your guidance in enhancing the clarity and accuracy of our analyses.

[Comment 3-4] Possible contamination: Were Suhum Municipality and Atiwa West District locations close with each other? Is it possible that people in Atiwa West District used maternal health services in Suhum Municipality or received part of the intervention package? The authors might want to expand explanations on the choice of the intervention and control group sites considering contamination possibilities. 

[Response 3-4] Thanks for sharing this concern. While fully understand reviewer’s concern about possible contamination, we made careful efforts to minimize the possibilities of contaminations by taking two approaches. First, we excluded those relocated from Suhum Municipality to Atiwa West District from the control group participants. Please see “Those having been relocated from Suhum Municipality and other six intervention municipalities/districts to Atiwa West District during their pregnancies were excluded from the follow-up sampling frame for the control group (Group C2 in Figure 1). By doing so, we ensured that, in the control group, the follow-up study participants were never, even partially, exposed to the intervention during their pregnancies” (Line 289-294 “Revised Manuscript with Track Changes”; lines ● of “Revised Manuscript”). Second, we took Atiwa West District which is not adjacent to Suhum Municipality (intervention group), as control group. East Akim District is located between Atiwa West district and Shum Municipality. The road distance from Suhum- East Akim border to East Akim- Atiwa West border is 65km. Thus, contamination is least likely to occur.

[Comment 3-5] Terminology: Is the word “difference-in-differences” more common than “difference-in-difference,” which the authors used? Please disregard this comment if the authors had a good reason to call their method difference-in-difference.

[Response 3-5] Thanks for your kind advice. It is really appreciated. Having fully agreed to the reviewer’s suggestion, now “difference-in-difference” has been replaced by “difference-in-differences” systematically in the manuscript. (see Line 107, Line 248, Line 296, Line 457, Line 465, Line 483, Line 1000, Figure 1 title, Figure 2 title, and Figure 3 title)

[Comment 3-6] Estimation of Wealth-group-specific difference-in-differences: The approach used for the analysis of heterogeneity in DID across different wealth groups is so-called difference-in-difference-in-differences, or DDD (for example, Olden and Møen [2022], https://academic.oup.com/ectj/article/25/3/531/6545797). According to Table 5, the authors’ specification may include the variables of group, time, wealth, and DDD (group*time*wealth). Compared to Equation 3.1 in Olden and Møen (2022), the model in Table 5 might miss the interaction terms group*wealth and time*wealth (the interaction term of time*group may be captured as “Adjusted DID estimates among the poor”). Consequently, DDD estimator in Table 5 might cause a bias by capturing two interaction terms (group*wealth and time*wealth), in addition to the appropriate DDD (group*time*wealth). Please check the model in Table 5 to ensure if it was specified appropriately and possibly re-estimate DDD if needed. 

[Response 3-6] We have carefully reviewed our model specification in light of your suggestions and the referenced article by Olden and Møen (2022). You correctly noted that our initial model omitted crucial interaction terms (group*wealth and time*wealth) that are necessary for accurately estimating the DDD approach. These terms are indeed essential for capturing the heterogeneity in effects across different wealth groups and time periods without introducing bias. In response to this, we have revised our model to include both group*wealth and time*wealth interaction terms. This amendment aligns our approach with the methodology outlined in Equation 3.1 of Olden and Møen (2022) and ensures the robustness of our DDD estimator by appropriately accounting for the multiple layers of interaction effects. The revised model and its results are now included in Table 5 of the manuscript. We believe that these changes have significantly strengthened our analysis. We highly appreciate your attention to detail and your guidance, which have been instrumental in enhancing the quality and accuracy of our work.

[Comment 3-7] Percent or percentage point change: In Figures 3 and 4, the authors presented crude DID as a percent form. However, it looks like it should have been “percentage points,” as this is the subtraction of percentages. Please find the following example for details: http://sumn.org/downloads/Percentage_Change.pdf.

[Response 3-7] Many thanks for sharing this very careful suggestion. Having agreed to this suggestion, the sentence “Crude DID is expressed in form of percent difference.” has been added to the footnotes of Figure 3 and Figure 4.

[Comment 3-8] PLoS authors have the option to publish the peer review history of their article. If published, this will include your full peer review and any attached files.

[Response 3-8] Thanks for your suggestion. All the coauthors agreed to publish the peer review history.

END

---

## [Decision Letter · Decision Letter 2]

26 Jul 2024

PONE-D-23-10669R2Effectiveness of a community-based intervention package in maternal health service utilizations: A cross-sectional quasi-experimental study in rural GhanaPLOS ONE

Dear Dr. Aiga,

Thank you for submitting your manuscript to PLOS ONE. After careful consideration, we feel that it has merit but does not fully meet PLOS ONE’s publication criteria as it currently stands. Therefore, we invite you to submit a revised version of the manuscript that addresses the points raised during the review process.

We look forward to receiving your revised manuscript.

Kind regards,

Mohammed Moinuddin, PhD

Academic Editor

PLOS ONE

Journal Requirements:

Reviewers' comments:

Reviewer's Responses to Questions

**Comments to the Author**

1. If the authors have adequately addressed your comments raised in a previous round of review and you feel that this manuscript is now acceptable for publication, you may indicate that here to bypass the “Comments to the Author” section, enter your conflict of interest statement in the “Confidential to Editor” section, and submit your "Accept" recommendation.

Reviewer #3: All comments have been addressed

Reviewer #4: All comments have been addressed

2. Is the manuscript technically sound, and do the data support the conclusions?

Reviewer #3: Yes

Reviewer #4: Yes

3. Has the statistical analysis been performed appropriately and rigorously? 

Reviewer #3: Yes

Reviewer #4: Yes

4. Have the authors made all data underlying the findings in their manuscript fully available?

Reviewer #3: Yes

Reviewer #4: No

5. Is the manuscript presented in an intelligible fashion and written in standard English?

Reviewer #3: Yes

Reviewer #4: Yes

6. Review Comments to the Author

Reviewer #3: The authors addressed most of the comments made in the previous round of peer review appropriately. Please consider the following remaining comments:

1. Design effect: In the responses-to-reviewers document, the authors explained that the design effect (1.3) used for sample size calculations was based on the Ghana DHS 2022 Report. However, the data collection started on May 2022, before the data collection of the Ghana DHS 2022 (starting on October 2022), the authors might not access to the information in the report when the survey was designed. In addition, it would be noted that 1) design effect is a function of Intraclass Correlation Coefficient (ICC) and cluster size (for example, https://doi.org/10.1016/j.jeph.2024.202198) and 2) cluster size may be different between DHS and this study as it depends on the sampling design of a study. Therefore, it is possible that the design effect expected in this study was larger than the design effect in DHS (if the [average] cluster size of this study is larger than the cluster size of DHS). It is unclear about the cluster size in this study (it should be reported if it is not mentioned in the Methods section), and the cluster size (or the number of samples in the second stage) of Ghana DHS 2022 was 30 according to its report (https://www.dhsprogram.com/pubs/pdf/FR387/FR387.pdf). Thus, if the average cluster size of this study was smaller than Ghana DHS, the authors’ sample size calculation might be sufficient (even if they knew it after the DHS report was issued). Otherwise, the authors might want to add explanations on their initial sample size calculations more in details in the Methods section and a possibility of a smaller sample size than it should have been as a limitation of this study in the Discussion section.

2. Percentage points: In Figures 2 and 3, the authors added the explanation that “Crude DID is expressed in form of percentage difference.” However, the DID may be captured as a difference between the percentages. For example, in the panel (a) of Figure 2, Crude DID was 4.6 percentage points (= [93.4% – 93.1%] – [92.4% – 96.7%] = 0.3% – [-4.3%]). Therefore, in the panel (a), it may be appropriate to present “Crude DID = 4.6 percentage points” instead of “Crude DID = 4.6%.” (If this is correct, the authors might want to revise similar presentations in Figures 2 and 3.

3. Percentage points: In relation to the comment above, the explanation of the Results section might be re-examined. For example, in Lines 487-, the authors explained that “Again, ANC was the only outcome variable that produced significant crude DID (11.5%; P = 0.082 < 0.10).” This percentage presentation (11.5%) might also be 11.5 percentage points. In addition, if it referred to the result presented in Figure 4 [1a] (ANC for women from poor 33%), it would be 11.4 percentage points and p = 0.085, according to the figure. Please check if the explanation in the main text was consistent with what was presented in the figure.

Reviewer #4: Introduction

1. “This is largely because they are not necessarily interactive and friendly enough to local omen” line 178. Who is this sentence referring to?

2. “Peer education has been often employed” – line 180 – grammar??

3. “a study needs to be conducted in Ghana”. Line 216 and 217 – this sentence should be re-stated. The study is done now, apparently.

4. “This is because the implementation of the Program was less affected by the pandemic of SARS-CoV-2 (COVID-19) pandemic in Ghana, compared with other three countries.” Line 218 – why? Unless this can be explained with evidence, the statement is more or less a conjecture and should be revised.

5. at least four ANC visits; (ii) FBD; and (iii) PNC. National Health Insurance Scheme (NHIS) line 218. Why use abbreviation in stating your objectives?

Methods

1. “Ghana was targeted for the study. i.e. Group I1 as the intervention group at baseline, Group I2 as the intervention group at follow-up, Group C1 as the control group at baseline, and Group as the control group at follow-up (Figure 1)”. – line 253. It does not clear to me whether baseline and follow up groups for both intervention and control had same of member for the time difference. Otherwise, how did you manage the difference in characteristics if each group had different participants?

2. “Atiwa West District was appropriate as the control group also because its socio-economic and socio-demographic characteristics, and maternal health service coverages at the baseline were reportedly at the similar level to those of Suhum Municipality.” – line 269, 270 – Atiwa West district is close to Suhum. They share border. How did you manage spillover effect?? If you did not, state it as a limitation.

Design and intervention

1. Ghana and indeed the WHO had abandoned 4 plus visit, even before 2018. What was the motivation for setting your outcome variable at 4+ instead of 8+?

Results

2. Data collection was conducted during the period from 5th to 29th May 2022, the final stage of the five-year Program – line 433. Authors may take this sentence away from results section and move it to methods section.

Discussion

“Of the three types of essential maternal health service utilizations (ANC, FBD and PPC), the intervention package improved only ANC visit coverage. A positive impact of the intervention package on FBD and PPC was not observed” – line 507 to 509 – it appears to me that analysing with 8+ plus in accordance WHO current recommendation even before this intervention was rolled out would have given different results altogether. Authors can re-run with 8+ or offer strong explanation n to support their choice of 4+

Overall, a clean statistical analysis.

7. PLOS authors have the option to publish the peer review history of their article (what does this mean?). If published, this will include your full peer review and any attached files.

Reviewer #3: No

Reviewer #4: No

---

## [Author Response · Author response to Decision Letter 2]

11 Aug 2024

RESPONSES TO THE REVIEWERS’ COMMENTS

We are extremely grateful to the editor and reviewers for their comments on the manuscript. We have revised the manuscript based on the useful comments. Please refer to the following point-by-point-based responses to the respective comments, along with revised manuscript. Please note that the changes in the revised manuscript (i.e. main text, tables/figures, and reference list) have been shown in: (i) red in this file for “Responses to Reviewers” as follows; (ii) track changes in the file “Revised Manuscript with Track Changes”; and (iii) red in the cleaned revised file “Revised Manuscript”.

COMMENTS FROM REVIEWER #3

[Comment 3-1] The authors addressed most of the comments made in the previous round of peer review appropriately. Please consider the following remaining comments:

[Response 3-1] Thanks for recognizing the responses and revision we have made for the previous version.

[Comment 3-2] 1. Design effect: In the responses-to-reviewers document, the authors explained that the design effect (1.3) used for sample size calculations was based on the Ghana DHS 2022 Report. However, the data collection started on May 2022, before the data collection of the Ghana DHS 2022 (starting on October 2022), the authors might not access to the information in the report when the survey was designed. In addition, it would be noted that 1) design effect is a function of Intraclass Correlation Coefficient (ICC) and cluster size (for example, https://doi.org/10.1016/j.jeph.2024.202198) and 2) cluster size may be different between DHS and this study as it depends on the sampling design of a study. Therefore, it is possible that the design effect expected in this study was larger than the design effect in DHS (if the [average] cluster size of this study is larger than the cluster size of DHS). It is unclear about the cluster size in this study (it should be reported if it is not mentioned in the Methods section), and the cluster size (or the number of samples in the second stage) of Ghana DHS 2022 was 30 according to its report (https://www.dhsprogram.com/pubs/pdf/FR387/FR387.pdf). Thus, if the average cluster size of this study was smaller than Ghana DHS, the authors’ sample size calculation might be sufficient (even if they knew it after the DHS report was issued). Otherwise, the authors might want to add explanations on their initial sample size calculations more in details in the Methods section and a possibility of a smaller sample size than it should have been as a limitation of this study in the Discussion section.

[Response 3-2] Thanks for sharing the detailed insights on design effect and two-stage sampling. As Reviewer #3 assumed, the design effect of 1.3 reported in Ghana DHS 2022 Report was accessed only after our data collection was completed. That was why this point and its reference (i.e. Ghana DHS 2022 Report) were not inserted into the main text when revising the manuscript last time. In view of this comment from Reviewer #3, the phrase “where mean cluster size was 25” has been inserted into the main text (see Line 373 in revised manuscript with TC; and Line 369 in revised manuscript without TC). We have reservation about adding more explanations on the sample size calculation in relation to design effect, for the three reasons: (i) the number of words of the main text has already great enough (i.e. 5,030) though PLoS One does not specify the word count limit; (ii) this study is not intended to focus on the detailed on sampling technique; and (iii) this manuscript has been submitted to PLoS One which is the journal not specialized in the journal specialized in epidemiology and/or biostatistics. 

[Comment 3-3] 2. Percentage points: In Figures 2 and 3, the authors added the explanation that “Crude DID is expressed in form of percentage difference.” However, the DID may be captured as a difference between the percentages. For example, in the panel (a) of Figure 2, Crude DID was 4.6 percentage points (= [93.4% – 93.1%] – [92.4% – 96.7%] = 0.3% – [-4.3%]). Therefore, in the panel (a), it may be appropriate to present “Crude DID = 4.6 percentage points” instead of “Crude DID = 4.6%.” (If this is correct, the authors might want to revise similar presentations in Figures 2 and 3.

[Response 3-3] Thanks for suggesting this correction. Accordingly, the phrase “Crude DID = X.XX percentage points” has been used for each figure in Figure 3 and Figure 4. Moreover, “Crude DID is expressed in form of percentage difference” has been replaced by “Crude DID is expressed in form of percentage points” (the footnotes for both Figure 3 and Figure 4 in the revised manuscript with TC and revised manuscript without TC).

[Comment 3-4] 3. Percentage points: In relation to the comment above, the explanation of the Results section might be re-examined. For example, in Lines 487-, the authors explained that “Again, ANC was the only outcome variable that produced significant crude DID (11.5%; P = 0.082 < 0.10).” This percentage presentation (11.5%) might also be 11.5 percentage points. In addition, if it referred to the result presented in Figure 4 [1a] (ANC for women from poor 33%), it would be 11.4 percentage points and p = 0.085, according to the figure. Please check if the explanation in the main text was consistent with what was presented in the figure.

[Response 3-4] Many thanks for suggesting this correction in line with the above [Comment 3-3]. Accordingly, “……significant crude DID (11.5 %; P = 0.082).” has been replaced by “……significant crude DID (11.4 percentage points; P = 0.085).” (see Line 500-501 in revised manuscript with TC; and Line 495-496 in revised manuscript without TC)

[Comment 3-5] Line 123-124: please consider writing "... the transition be smooth by increasing the number of ANC visits."

[Response 3-5] Thanks for identifying this typo by carefully reviewing the revised manuscript. Accordingly, the phrase "... the transition be smooth by increasing in the number of ANC visits." has been replaced by "... the transition be smooth by increasing the number of ANC visits." (see Line 122-123 in revised manuscript with TC; and Line 122-123 in revised manuscript without TC).

COMMENTS FROM REVIEWER #4

[Comment 4-1] Introduction 1. “This is largely because they are not necessarily interactive and friendly enough to local women” line 178. Who is this sentence referring to?

[Response 4-1] Thanks for this clarification. Accordingly, “This is largely because they are not necessarily interactive and friendly enough to local women” has been replaced by “This is largely because the formal health workers' way of demand creation is not necessarily interactive and friendly enough to local women” (see Line 177 in revised manuscript with TC; and Line 177 in revised manuscript without TC). Hope this will help Reviewer #4 and the audience of the article will more clearly understand this part.

[Comment 4-2] Introduction 2. “Peer education has been often employed” – line 180 – grammar??

[Response 4-2] Thanks for this clarification. This way of using the verb “employ” is common. For example, “The peer education program has been employed as one of the strategies for HIV prevention” (Hayashi M, Evaluation of the Peer Education Program in the Central Region of Vietnam. Journal of the National Institute of Public Health. 2009; 58 (4): 408) and “Peer education interventions are commonly employed to prevent HIV” (Abdi F, Simbar M. The Peer Education Approach in Adolescents- Narrative Review Article. Iran J Public Health. 2013; 42 (11): 1200-1206). Many other earlier publications used “employ” in this manner. Moreover, in fact, none of the three other reviewers for our manuscript raised this clarification. This implies the sentence “Peer education has been often employed” is well understood. Thus, no change has been made in the manuscript. 

[Comment 4-3] Introduction 3. “a study needs to be conducted in Ghana”. Line 216 and 217 – this sentence should be re-stated. The study is done now, apparently.

[Response 4-3] Thanks for this suggestion. We agreed to make a necessary change in this sentence. Now, “… a study needs to be conducted in Ghana” has been replaced by “… this study was conducted in Ghana” (see Line 220 in revised manuscript with TC; and Line 217 in revised manuscript without TC).

[Comment 4-4] Introduction 4. “This is because the implementation of the Program was less affected by the pandemic of SARS-CoV-2 (COVID-19) pandemic in Ghana, compared with other three countries.” Line 218 – why? Unless this can be explained with evidence, the statement is more or less a conjecture and should be revised.

[Response 4-4] Thanks for this clarification. There is some evidence for the phrase “… the Program was less affected by the SARS-CoV-2 (COVID-19) pandemic in Ghana” compared with three other countries. First, the Ghanaian government established partial lockdowns in COVID-19 hotspots, primarily in Greater Accra Region and Greater Kumasi Region on 30th March 2020. Thus, Eastern Region in which both Suhum Municipality (intervention group) and Atiwa West District (control group), the target areas of this study, were not included in the partial lockdown areas. Second, Ghana was one of the first African countries that lifted the lockdowns (Knott S. Ghana’s decision to lift partial COVID lockdown criticized by some. VOA News. April 20, 2020. https://www.voanews.com/a/africa_ghanas-decision-lift-partial-covid-19-lockdown-criticized-some/6187869.html). 

[Comment 4-5] Introduction 5. at least four ANC visits; (ii) FBD; and (iii) PNC. National Health Insurance Scheme (NHIS) line 218. Why use abbreviation in stating your objectives?

[Response 4-5] Thanks for clarification. First, “PNC” was the typo of “PPC”. Second, the three abbreviations “ANC”, “FBD” and “PPC” have appeared before this part in Line 160-162. I.e. “To reduce a significantly greater number of maternal deaths in Sub-Saharan Africa, timely and quality antenatal care (ANC) and facility-based delivery (FBD), post-partum care (PPC) ….” (see Line 160-162 in revised manuscript with TC; and Line 160-162 in revised manuscript without TC). Fully spelling out an abbreviation should be done only once, when it appears for the first time. Thus, no change has been made in Line 235-236.

[Comment 4-6] Methods 1. “Ghana was targeted for the study. i.e. Group I1 as the intervention group at baseline, Group I2 as the intervention group at follow-up, Group C1 as the control group at baseline, and Group as the control group at follow-up (Figure 1)”. – line 253. It does not clear to me whether baseline and follow up groups for both intervention and control had same of member for the time difference. Otherwise, how did you manage the difference in characteristics if each group had different participants?

[Response 4-6] Thanks for this clarification. Needless to say, the women who were pregnant at the time of the baseline (from 1st January 2018 to 28th February 2019) could be generally expected not to be pregnant again at the time of the follow-up (from 1st August 2021 to 30th April 2022), too. Thus, the study participants in baseline and follow-up groups were different. Yet, there were a few possibilities that some women having been pregnant at the time of both baseline and follow-up. For this reason, we systematically checked the presence of those women. At the result, it was found that no women who were pregnant at the time of both baseline and follow-up were selected for any of the four groups. Please refer to the two sentences “All the women who delivered between 1st January 2018 and 28th February 2019 before the Program’s launch in March 2019 were included in the baseline sampling frames for both intervention and control groups (Group I1 and Group C1 in Figure 1). Similarly, all the women who delivered between 1st August 2021 and 30th April 2022 after the 10 months had passed since the Program’s launch were included in the follow-up sampling frames for both intervention and control groups (Group I2 and Group C2 in Figure 1).” (see Line 289-295 in revised manuscript with TC; and Line 282-288 in the revised manuscript without TC).

[Comment 4-7] Methods 2. “Atiwa West District was appropriate as the control group also because its socio-economic and socio-demographic characteristics, and maternal health service coverages at the baseline were reportedly at the similar level to those of Suhum Municipality.” – line 269, 270 – Atiwa West district is close to Suhum. They share border. How did you manage spillover effect?? If you did not, state it as a limitation.

[Response 4-7] Thanks for this clarification. The possibilities of contamination of a series of interventions into Atiwa West District (control group) must have been limited for the two reasons. First, Atiwa West District (control group) is closely located to Suhum Municipality (intervention group) but without sharing their boarder (see the map below). Since the total number of tables and figures is too many (i.e. 9 = 5 tables and 4 figures), we decided not to add this map to the revised manuscript. 

Second, during COVID-19 pandemic, people’s mobility and internal migration within Ghana were regulated and refrained from. For example, the graph below (https://www.exemplars.health/emerging-topics/ecr/ghana/how-did-ghana-respond) shows the reduction in people’s movements during the COVID-19 pandemic. Nevertheless, we cannot negate some possibilities of contamination. Therefore, an additional paragraph has been inserted into the final part of Discussion section. I.e. “Fourth, there are limited possibilities of contamination of a series of interventions into the control group. This is because: (i) Atiwa West District (control group) and Suhum Municipality (intervention group) are not mutually adjacent; and (ii) people’s movement was regulated and voluntarily refrained from during COVID-19 pandemic. Yet, a certain level of the possibilities of contamination cannot be negated.” (see Line 648-652 in revised manuscript with TC; and Line 638-642 in the revised manuscript without TC).

[Comment 4-8] Design and intervention 1. Ghana and indeed the WHO had abandoned 4 plus visit, even before 2018. What was the motivation for setting your outcome variable at 4+ instead of 8+?

Results

[Response 4-8] Thanks for raising this point. As Reviewer #4 mentioned, Ghana Health Service (GHS) recommended all pregnant women make at least eight ANC visits in Ghana National Safe Motherhood Protocol 2017. Yet, in view of this unrealistic and infeasible target of eight or more NC visits, GHS relaxed the target figure for the minimum number of ANC visits from eight back to four in 2020. In fact, Ghana National Safe Motherhood Protocol 2021 employs at least four ANC visits as the minimum number of ANC visits (GHS, Ghana National Safe Motherhood Protocol 2021. Accra: GHS; 2021. page 3) 4+ANC has been serving as the only monitoring indicator of frequency of ANC visits in GHS (GHS, Health Information Management System. Standard Operating Procedure 4th edition 2020. Accra: GHS; 2017. page 337). Even, Ghana DHS 2022 employed exclusively at least four ANC visits by excluding at least eight ANC visits as the variable, through respecting this GHS’s relaxing the national standard (Ghana Statistical Service, Ghana Demographic and Health Survey 2022. Accra: Ghana Statistical Service; 2022. Page 158.) To avoid unnecessary complicated explanation which is not essential to this study, we decided not to insert any additional sentences and phrases on this topic into the main text.

[Comment 4-9] final stage of the five-year Program – line 433. Authors may take this sentence away from results section and move it to methods section.

[Response 4-9] Thanks for this suggestion. The phrase “, the final stage of the five-year Program” has been simply deleted without moving it to Methods section. This is because this phrase is essential neither in Results section nor in Methods section. (see Line 445 in revised manuscript with TC; and Line 438 in revised manuscript without TC).

[Comment 4-10] Discussion: “Of th

---

## [Decision Letter · Decision Letter 3]

29 Sep 2024

Effectiveness of a community-based intervention package in maternal health service utilizations: A cross-sectional quasi-experimental study in rural Ghana

PONE-D-23-10669R3

Dear Dr. Aiga,

We’re pleased to inform you that your manuscript has been judged scientifically suitable for publication and will be formally accepted for publication once it meets all outstanding technical requirements.

Kind regards,

Mohammed Moinuddin, PhD

Academic Editor

PLOS ONE

Additional Editor Comments (optional):

Reviewers' comments:

Reviewer's Responses to Questions

**Comments to the Author**

1. If the authors have adequately addressed your comments raised in a previous round of review and you feel that this manuscript is now acceptable for publication, you may indicate that here to bypass the “Comments to the Author” section, enter your conflict of interest statement in the “Confidential to Editor” section, and submit your "Accept" recommendation.

Reviewer #3: All comments have been addressed

2. Is the manuscript technically sound, and do the data support the conclusions?

Reviewer #3: Yes

3. Has the statistical analysis been performed appropriately and rigorously? 

Reviewer #3: Yes

4. Have the authors made all data underlying the findings in their manuscript fully available?

Reviewer #3: Yes

5. Is the manuscript presented in an intelligible fashion and written in standard English?

Reviewer #3: Yes

6. Review Comments to the Author

Reviewer #3: The authors addressed all the comments made in the previous round of peer-review, particularly regarding the design effect issue.

7. PLOS authors have the option to publish the peer review history of their article (what does this mean?). If published, this will include your full peer review and any attached files.

Reviewer #3: **Yes: **Akira Shibanuma

---

## [Editor Report · Acceptance letter]

25 Oct 2024

PONE-D-23-10669R3 

PLOS ONE

Dear Dr. Aiga, 

I'm pleased to inform you that your manuscript has been deemed suitable for publication in PLOS ONE. Congratulations! Your manuscript is now being handed over to our production team.

Kind regards, 

on behalf of

Dr Mohammed Moinuddin 

Academic Editor

PLOS ONE